EMBO
Molecular Medicine

# Reduction in mitochondrial iron alleviates cardiac damage during injury

Hsiang-Chun Chang[†,1], Rongxue Wu[†,1], Meng Shang[1], Tatsuya Sato[1], Chunlei Chen[1], Jason S Shapiro[1], Ting Liu[1], Anita Thakur[1], Konrad T Sawicki[1], Sathyamangla VN Prasad[2] & Hossein Ardehali[1,*]

## Abstract

**Excess cellular iron increases reactive oxygen species (ROS) production and causes cellular damage. Mitochondria are the major site of iron metabolism and ROS production; however, few studies have investigated the role of mitochondrial iron in the development of cardiac disorders, such as ischemic heart disease or cardiomyopathy (CM). We observe increased mitochondrial iron in mice after ischemia/reperfusion (I/R) and in human hearts with ischemic CM, and hypothesize that decreasing mitochondrial iron protects against I/R damage and the development of CM. Reducing mitochondrial iron genetically through cardiac-specific overexpression of a mitochondrial iron export protein or pharmacologically using a mitochondria-permeable iron chelator protects mice against I/R injury. Furthermore, decreasing mitochondrial iron protects the murine hearts in a model of spontaneous CM with mitochondrial iron accumulation. Reduced mitochondrial ROS that is independent of alterations in the electron transport chain's ROS producing capacity contributes to the protective effects. Overall, our findings suggest that mitochondrial iron contributes to cardiac ischemic damage, and may be a novel therapeutic target against ischemic heart disease.**

**Keywords** heart failure; iron; ischemia; ischemia/reperfusion; mitochondria
**Subject Categories** Cardiovascular System; Metabolism

## Introduction

Cardiovascular disease accounts for nearly six hundred thousand deaths per year in United States (Heron, 2013) and over 4 million deaths in Europe (Nichols *et al*, 2012), making it the most common cause of death in the Western World. While current therapies improve survival from the initial myocardial infarction (Simoons *et al*, 1986; Hollenbeck *et al*, 2014), many patients ultimately develop heart failure (Liang & Delehanty, 2009), which poses great

financial burden on the healthcare system (Braunwald, 2013) and significantly diminishes quality of life. It is believed that the extent of cardiac tissue damage is correlated with the development of heart failure (Foo *et al*, 2005; McAlindon *et al*, 2015); however, no clinically available therapy directly targets cardiomyocytes in order to reduce damage after ischemia/reperfusion (I/R) injury. Therefore, the development of novel therapies targeting cardiomyocyte death is essential.

Iron is a required element for normal cellular processes, including cellular respiration (Gille & Reichmann, 2011; Hirst, 2013), protein production (Kispal *et al*, 2005), lipid metabolism (Shakoury-Elizeh *et al*, 2010), and DNA replication (Furukawa *et al*, 1992). Normally, iron is used for heme and iron/sulfur (Fe/S) cluster synthesis in mitochondria or is stored in ferritin molecules in the cytoplasm (De Domenico *et al*, 2008; Hentze *et al*, 2010; Ye & Rouault, 2010) or in mitochondrial ferritin molecules in mitochondria (Napier *et al*, 2005; Horowitz & Greenamyre, 2010; Vigani *et al*, 2013; Li *et al*, 2014). Excess iron can cause tissue damage through the production of reactive oxygen species (ROS) via the Fenton-like and Harbor–Weiss reactions (Aigner *et al*, 2008). Iron-catalyzed ROS formation can also increase free iron through reactions with Fe/S clusters or other forms of loosely bound iron (Sideri *et al*, 2009; Gomez *et al*, 2014). Previous studies have demonstrated a role for lysosomal iron in radiation-mediated or $H_2O_2$-induced cell death (Yu *et al*, 2003; Persson *et al*, 2005; Kurz *et al*, 2010). Additionally, in diseases with mitochondrial iron overload, iron-mediated mitochondrial DNA and membrane damage have been linked to mitochondrial dysfunction (Eaton & Qian, 2002; Gao *et al*, 2009). Increased iron has been described in the setting of I/R in various organs (Zhao *et al*, 1997; Comporti *et al*, 2002; Ghio, 2009; Kaushal & Shah, 2014), and increased transferrin receptor 1 expression due to activation of hypoxia inducible factor signaling has been implicated as a mechanism for the change in cellular iron (Tang *et al*, 2008). While one study suggested that extracellular iron is involved in renal I/R injury (de Vries *et al*, 2004), very few studies have discerned the contribution of baseline iron in various subcellular compartments to I/R injury in cells and animals.

Because of iron's contribution to ROS production, several studies have evaluated the efficacy of iron chelation in alleviating tissue

1 Feinberg Cardiovascular Research Institute (FCVRI), Northwestern University Feinberg School of Medicine, Chicago, IL, USA
2 Department of Molecular Cardiology, Lerner Research Institute, Cleveland Clinic Foundation, Cleveland, OH, USA
*Corresponding author. Tel: +1 312 503 2342; Fax: +1 312 503 0137; E-mail: h-ardehali@northwestern.edu
†These authors contributed equally to this work

damage during myocardial infarction, but the results have been controversial. Deferoxamine (DFO), a Federal Food and Drug Administration approved iron chelator for transfusion-related iron overload, has a strong affinity for iron but low cellular permeability. DFO treatment was shown to improve cardiac function after I/R in an *ex vivo* heart perfusion system (Badylak *et al*, 1987; Williams *et al*, 1991; Nicholson *et al*, 1997), and in canine and porcine models of I/R *in vivo* (Ramesh Reddy *et al*, 1989; Lesnefsky *et al*, 1990b; Kobayashi *et al*, 1991; Chopra *et al*, 1992). Infusion of DFO was also associated with improved cardiac function in patients who had undergone coronary artery bypass surgery (Menasche *et al*, 1990; Paraskevaidis *et al*, 2005), and iron chelation with DFO in patients with thalassemia major led to improved cardiac function and survival (Marcus *et al*, 1984; Pepe *et al*, 2011; Porter *et al*, 2013). On the other hand, some reports using DFO in large animal models or primate models of I/R failed to reduce the infarct size after injury (Lesnefsky *et al*, 1990a; Watanabe *et al*, 1993; Chatziathanasiou *et al*, 2012). Tissue penetrance was cited as a potential cause of the lack of protective effects. This explanation is further supported by the ability of another cell-permeable iron chelator, 2,2′-bipyridyl (BPD), to protect rats against cerebral infarction (Demougeot *et al*, 2004; Méthy *et al*, 2008; Wu *et al*, 2012). It should also be noted that these studies all assessed the effects of iron chelation on total cellular iron, and did not distinguish iron in different subcellular compartments.

Taken together, these studies indicate that differential cellular iron localization in the heart may have functional consequences in cardiovascular disease. In the current paper, we first observe an increase in mitochondrial iron after cardiac I/R injury in mice and in cardiac tissue samples from patients with ischemic cardiomyopathy (CM) compared to non-failing hearts, which leads us to hypothesize that reducing baseline mitochondrial iron would protect the heart against I/R injury and the development of CM. We show that pharmacologic reduction in baseline mitochondrial iron, but not cytoplasmic iron, protects cells against $H_2O_2$-induced cell death. Our *in vivo* data with two distinct approaches of mitochondrial iron modulation clearly indicate that a decrease in baseline mitochondrial iron is protective against cardiac I/R injury. Importantly, mice with a modest decrease in cardiac mitochondrial iron display a normal phenotype at baseline. We also demonstrate that pharmacological reduction in mitochondrial iron prevents the development of cardiomyopathy in a genetic model of mitochondrial iron overload, thus providing clinical relevance for targeting mitochondrial iron. The protective effects of reducing mitochondrial iron in both disease models are associated with reduced ROS production during injury.

# Results

### Mitochondrial non-heme iron increases after I/R injury and in human samples with ischemic cardiomyopathy

To investigate the acute changes in iron content in different subcellular localizations after I/R injury, we subjected wild-type C57/BL6 mice to I/R and measured cytoplasmic and mitochondrial non-heme iron in the hearts of mice 2 days after I/R. We first verified the purity of the subcellular fractions (Appendix Fig S1A).

While no significant changes in cytoplasmic non-heme iron were observed (Fig 1A), mitochondrial non-heme iron was significantly increased after I/R injury (Fig 1B). Since labile iron can catalyze the formation of ROS, which in turn further increases free iron, we measured chelatable mitochondrial and cytoplasmic iron in H9c2 cardiomyoblasts exposed to $H_2O_2$, a model designed to simulate the surge of ROS during the reperfusion stage of I/R. The treatment of $H_2O_2$ for 6 h significantly increased mitochondrial chelatable iron as well as cytoplasmic chelatable iron (Fig 1C and D). To put these findings into a clinical context, we measured mitochondrial and cytosolic non-heme iron in cardiac tissue samples from patients without heart failure and with ischemic cardiomyopathy (ISCM). Western blotting results demonstrated the purity of subcellular fractions (Appendix Fig S1B). Mitochondrial fractions from ISCM samples had a significantly higher level of non-heme iron, while no significant difference was observed in cytosolic non-heme iron between non-failing and ISCM heart samples (Fig 1E and F). These findings together suggest that mitochondrial non-heme iron increases after I/R and may participate in tissue injury.

### A decrease in baseline mitochondrial iron protects cardiomyocytes against $H_2O_2$-induced cell death *in vitro*

Our findings that mitochondrial non-heme iron is increased in the hearts of mice after I/R and failing human hearts and previous observations that labile iron can catalyze the conversion of hydrogen peroxide to the hydroxyl radical, a major source of tissue damage during I/R (Zweier & Talukder, 2006), prompted us to hypothesize that modulation of mitochondrial iron may protect against I/R injury. We chose two iron chelators with distinct mitochondrial permeability—DFO, which has poor penetrance through the cell membrane, and BPD, which has high membrane permeability and thus is able to access mitochondria (Demougeot *et al*, 2004). The dose of DFO treatment was based on previous published reports (Ichikawa *et al*, 2014). Various doses of BPD were tested for its ability to reduce cellular and mitochondrial iron (Appendix Fig S2). Based on the changes in mitochondrial iron levels in cells treated with various doses of BPD, we used 100 μM of BPD for subsequent *in vitro* studies. DFO and BPD caused significant decreases in both cytosolic and nuclear iron (Fig 2A and B); however, 2-h pre-treatment of H9c2 cardiomyoblasts with BPD, but not DFO, decreased mitochondrial labile iron (Fig 2C and D). While pre-treatment of H9c2 with DFO only conferred a slight protection against oxidative stress, BPD pre-treatment significantly reversed $H_2O_2$-induced cell death (Fig 2E). Additionally, the increase in mitochondrial labile iron after $H_2O_2$ treatment was attenuated by BPD but not by DFO pre-treatment (Fig 2F). Therefore, BPD, which can reduce mitochondrial iron, exerts protection against oxidative damage to the cell.

### Overexpression of ABCB8 in cardiomyocytes *in vivo* reduces mitochondrial iron and protects against I/R damage

Since modulation of mitochondrial iron protected cells against $H_2O_2$-induced cell death, we then tested whether similar protective effects could be observed *in vivo* using a cardiac I/R injury model. Previous *in vitro* studies demonstrated that overexpression of ABCB8, a protein found to be involved in mitochondrial iron export,

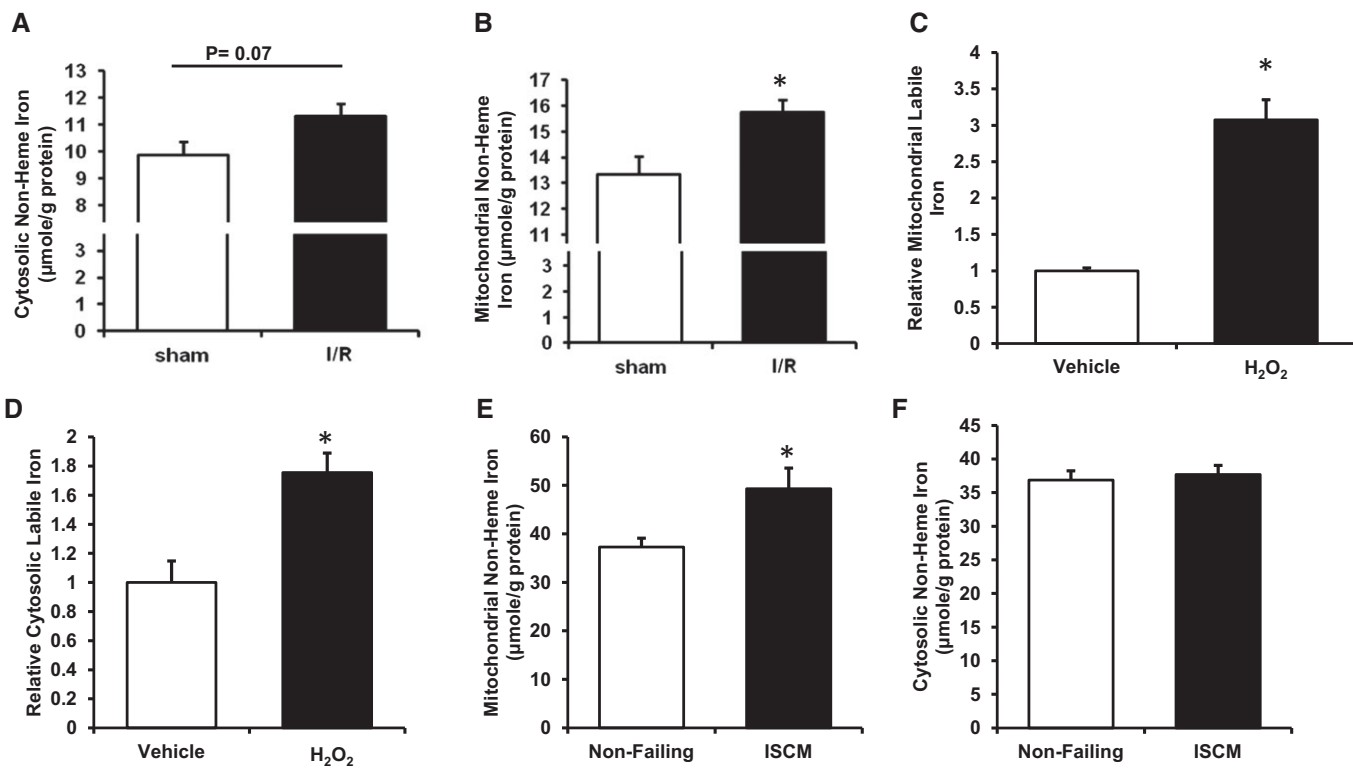

**Figure 1. Ischemia/reperfusion (I/R) injury *in vivo* causes increased mitochondrial iron.**

A   Cytosolic non-heme iron levels in wild-type mice subjected to sham or I/R procedure 2 days after surgery. Two-tailed unpaired *t*-test was performed. *N* = 4 mice for each group.

B   Mitochondrial non-heme iron levels in wild-type mice subjected to sham or I/R procedure 2 days after surgery. *P = 0.024 with two-tailed unpaired *t*-test. *N* = 4 mice for each group.

C   Mitochondrial labile iron in H9c2 cells with or without $H_2O_2$ treatment measured using RPA fluorescence. *P < 0.0001 with two-tailed unpaired *t*-test. *N* = 8 independent samples for PBS group and *N* = 10 independent samples for $H_2O_2$ group.

D   Cytosolic labile iron in H9c2 cells with or without $H_2O_2$ treatment measured using calcein fluorescence. *P = 0.0042 with two-tailed unpaired *t*-test. *N* = 11 independent samples in each group.

E   Mitochondrial iron in human cardiac tissue sample from non-failing hearts and from hearts with ischemic cardiomyopathy (ISCM). *P = 0.041 with two-tailed unpaired *t*-test. *N* = 4 independent samples in each group.

F   Cytosolic iron in human cardiac tissue sample from non-failing hearts and from hearts with ischemic cardiomyopathy (ISCM). Two-tailed unpaired *t*-test was performed. *N* = 4 independent samples in each group.

Data information: All data are expressed as mean ± SEM.

decreases mitochondrial iron (Ichikawa *et al*, 2012). We therefore used cardiac-specific overexpression of human ABCB8 driven by α-MHC promoter as a genetic model for decreased mitochondrial iron. We first verified overexpression of ABCB8 using quantitative real-time PCR against human ABCB8 and Western blotting (Appendix Fig S3A and B) and colocalization with mitochondria using confocal microscopy (Appendix Fig S3C). Consistent with the *in vitro* findings, the hearts of ABCB8 transgenic (TG) mice displayed lower mitochondrial non-heme iron levels at baseline without a significant difference in cytosolic non-heme iron levels (Fig 3A and B). ABCB8 TG mice demonstrated normal cardiac function at baseline (Fig 3C and D) and no change in ROS production (Fig 3E). In addition, the expression of key antioxidant systems, including *Sod1* and *Sod2,* in TG mice was similar to non-transgenic (NTG) littermates (Fig 3F).

We then subjected age-matched ABCB8 TG and NTG littermate mice to I/R or sham operation and monitored their cardiac function

with serial echocardiography. No difference in cardiac function was observed in mice receiving the sham operation, while ABCB8 TG mice displayed significantly better cardiac function compared to NTG littermates after I/R (Fig 4A and Appendix Fig S4). Two days after I/R, cell death was significantly lower in ABCB8 TG mice compared to NTG littermates (Fig 4B and C). Also, compared to NTG mice, ABCB8 TG mice demonstrated less expression of cardiac stress markers atrial natriuretic factor (*Nppa*), brain natriuretic peptide (*Nppb*), and β-myosin heavy chain (*Myh7*) during the acute phase of I/R injury (Fig 4D–F). Lastly, hematoxylin and eosin staining in the peri-infarct zone revealed reduced cellular damage in ABCB8 TG mice compared to NTG mice (Fig 5A). These results are consistent with reduced cardiac stress in ABCB8 TG mice compared to NTG mice after I/R injury.

We also studied the long-term remodeling of cardiac tissues after I/R, which was used as a proxy for estimating initial cellular damage. At 28 days after I/R, ABCB8 TG mice demonstrated

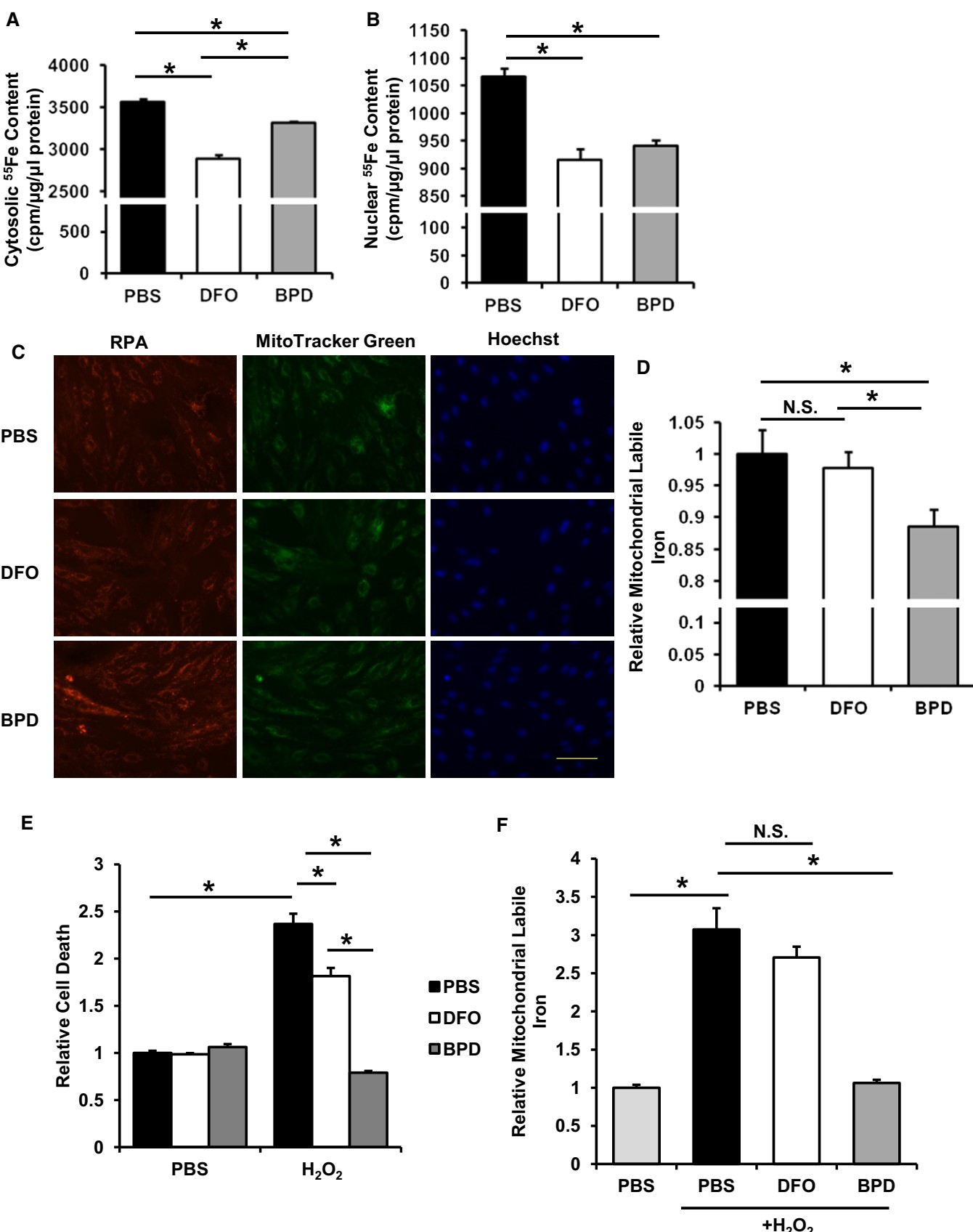

Figure 2.

**Figure 2.  Mitochondrial-permeable iron chelator is protective against oxidative stress *in vitro*.**

A   Cytosolic iron in H9c2 cells preloaded with radioactive $^{55}$Fe and treated with the indicated iron chelators. ANOVA followed by *post hoc* Tukey's test was performed. *P = 7.9E-8 PBS vs. DFO. *P = 9.4E-6 BPD vs. DFO. *P = 0.003 PBS vs. BPD. N = 4 independent samples for PBS group and N = 6 independent samples for other groups.

B   Nuclear iron in H9c2 cells preloaded with radioactive $^{55}$Fe and treated with the indicated iron chelators. ANOVA followed by *post hoc* Tukey's test was performed. *P = 0.0003 PBS vs. BPD. *P = 5.58E-5 PBS vs. DFO. N = 4 independent samples for PBS group and N = 6 independent samples for other groups.

C   Representative RPA fluorescence staining for labile mitochondrial iron in H9c2 cells with the indicated iron chelator treatment. Scale bar, 100 μm.

D   Labile mitochondrial iron measured by RPA fluorescence in H9c2 cells with the indicated iron chelator treatment. ANOVA followed by *post hoc* Tukey's test was performed.*P = 0.016 PBS vs. BPD. *P = 0.04 DFO vs. BPD. N = 8 independent samples for each group.

E   $H_2O_2$-induced cell death in H9c2 cells with the indicated treatments. ANOVA followed by *post hoc* Tukey's test was performed. *P = 5E-8 PBS-PBS vs. PBS-$H_2O_2$. *P = 4E-6 PBS-$H_2O_2$ vs. DFO-$H_2O_2$. *P = 5E-8 PBS-$H_2O_2$ vs. BPD-$H_2O_2$. *P = 1E-7 BPD-$H_2O_2$ vs. DFO-$H_2O_2$. N = 6 independent samples for each group.

F   Labile mitochondrial iron in H9c2 cells with the indicated treatment. PBS with and without $H_2O_2$ data was copied from Fig 1C. ANOVA followed by *post hoc* Tukey's test was performed. *P = 1E-8 PBS-PBS vs. PBS-$H_2O_2$. *P = 3E-9 PBS-$H_2O_2$ vs. BPD-$H_2O_2$. N = 8 independent samples for PBS-PBS and N = 12 for the other groups.

Data information: All data are expressed as mean ± SEM. N.S., not significant.

significantly lower fibrosis compared to their NTG littermates (Fig 5B and C). In addition, ABCB8 TG mice maintained reduced expression of *Nppa*, *Nppb,* and *Myh7* at the same time point (Fig 5D–F), consistent with reduced cardiac stress from the initial injury. These findings further support our hypothesis that decreasing baseline mitochondrial iron in cardiomyocytes is sufficient to protect against I/R injury.

**Pharmacological modulation of mitochondrial iron *in vivo* protects against I/R damage**

Our findings demonstrate that a decrease in mitochondrial iron (using a genetic mouse model) is protective against cardiac I/R injury. To provide more clinical applicability to our findings, we used the two iron chelators (DFO and BPD) from our *in vitro* studies and injected them into wild-type C57/BL6 mice prior to I/R to evaluate whether a pharmacological decrease in mitochondrial iron is sufficient to protect against I/R injury. After a 1-week regimen, BPD treatment (80 mg/kg per day) in mice at baseline decreased cardiac mitochondrial iron, while DFO treatment (50 mg/kg every other day) did not alter cardiac mitochondrial iron levels (Fig 6A). This finding is similar to our *in vitro* results. The inability of DFO to modulate mitochondrial iron was not due to its inactivity, as both treatments lowered cardiac cytosolic iron (Fig 6B). A decrease in cardiac nuclear iron was also observed but did not reach statistical significance with either of the iron chelators (Fig 6C), which can be due to relatively low amounts of iron in the nucleus (as seen comparing Fig 2A and B) and the colorimetric measurement having lower sensitivity compared to the radioactive measurement. The purity of the nuclear fraction was verified with Western blotting (Appendix Fig S1C). This finding is consistent with DFO being a strong iron chelator but having poor penetrance into mitochondria. Both drugs did not cause damage to the heart at baseline as assessed by cardiac ejection fraction and fractional shortening using echocardiography (Fig 6D and Appendix Fig S5A).

To test the use of these iron chelators in the setting of I/R, mice were pre-treated for a week with these chelators and then subjected to I/R or sham operation. Chelation was continued for 2 weeks after the operation. Mice treated with BPD had preserved cardiac function after I/R, while DFO failed to protect mice against I/R damage (Fig 6E and Appendix Fig S5B). BPD-treated mice

demonstrated less cardiac remodeling compared to either vehicle- or DFO-treated mice, which is consistent with milder cardiac damage during I/R (Fig 6F). Additionally, BPD but not DFO treatment attenuated the expression of *Nppa*, *Nppb,* and *Myh7* (Fig 6G–I). To rule out any changes of serum iron hematopoiesis that can account for the cardiac functional difference, we performed complete blood counts and measured serum iron parameters in wild-type mice treated with iron chelators for 3 weeks. As expected, the 3-week iron chelation regimen resulted in lower serum iron, but did not alter erythropoiesis as evidenced by comparable RBC count and hemoglobin level (Appendix Table S1). These observations suggest that pharmacologically lowering mitochondrial iron levels in the setting of I/R results in protective effects.

**A decrease in mitochondrial iron at baseline protects against the development of spontaneous cardiomyopathy in cardiac-specific ABCB8 knockout mice**

The above studies demonstrated that a reduction in baseline mitochondrial iron is protective against I/R injury. We next took a loss-of-function approach and studied whether a decrease in mitochondrial iron in cardiac-specific ABCB8 knockout mice, a model of spontaneous cardiomyopathy with mitochondrial iron overload (Ichikawa *et al*, 2012), protects against cardiac tissue damage. Because our above studies indicated that DFO does not modulate mitochondrial iron and ABCB8 knockout mice did not show cytoplasmic iron accumulation, we exclusively used BPD in the subsequent experiments. Mitochondrial iron in ABCB8 flox/flox (LoxP sequences flanking exon one of both alleles of ABCB8) mice with or without the α-MHC-MER-Cre-MER transgene was modulated with BPD or vehicle control for 1 week before the induction of ABCB8 deletion with tamoxifen. BPD treatment continued until 4 weeks after the completion of tamoxifen treatment, at which point mitochondrial iron and cardiac function were measured. The effectiveness of Cre-mediated gene excision was confirmed at the protein level (Appendix Fig S6). Cre$^+$ mice (ABCB8 knockout, KO) receiving vehicle treatment demonstrated mitochondrial iron accumulation, while BPD treatment prevented the development of iron accumulation. BPD treatment also resulted in a decrease in mitochondrial iron in Cre$^-$ mice (WT, Fig 7A). BPD treatment in ABCB8 KO mice

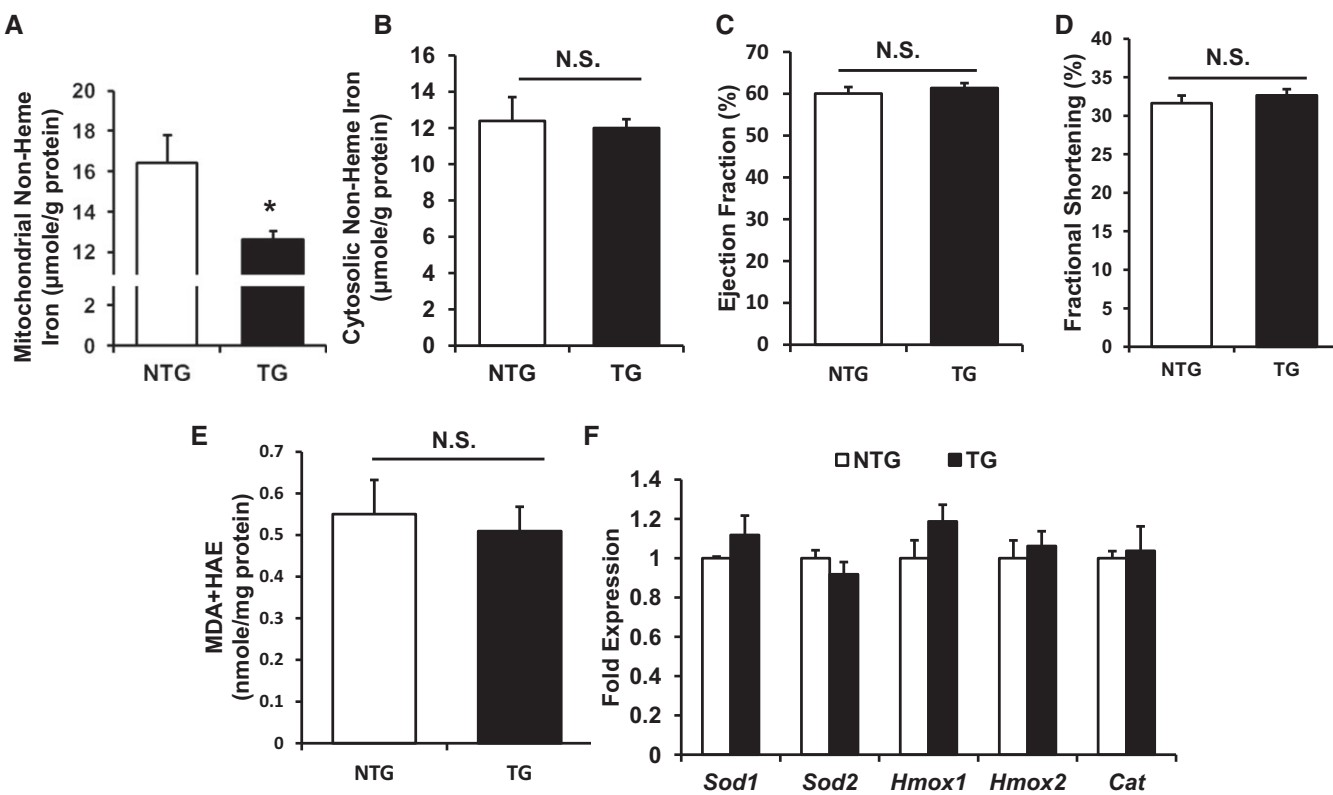

**Figure 3.  Cardiac-specific ABCB8 overexpression reduces mitochondrial iron but does not alter cardiac function, ROS or the antioxidant system.**

A   Mitochondrial non-heme iron in ABCB8 NTG and TG mice. $N$ = 4 mice for NTG and 6 mice for TG. *$P$ = 0.021 with two-tailed unpaired $t$-test.
B   Cytosolic non-heme iron in ABCB8 NTG and TG mice. $N$ = 4 mice for NTG and 6 mice for TG. Two-tailed unpaired $t$-test was performed.
C   Baseline ejection fraction of littermate ABCB8 NTG and TG mice. $N$ = 6 mice in each group. Two-tailed unpaired $t$-test was performed.
D   Baseline fractional shortening of littermate ABCB8 NTG and TG mice. $N$ = 6 mice in each group. Two-tailed unpaired $t$-test was performed.
E   Lipid peroxidation products in hearts of TG and NTG mice. $N$ = 6 mice in each group. Two-tailed unpaired $t$-test was performed.
F   Relative expression of antioxidant genes in NTG and TG mice. $N$ = 6 mice in each group. Two-tailed unpaired $t$-test was performed.

Data information: All data are expressed as mean ± SEM. N.S., not significant.

preserved cardiac function, while the treatment itself was not cardiotoxic (Fig 7B and C). Additionally, BPD-treated ABCB8 KO mice demonstrated attenuated expression of *Nppa*, *Nppb,* and *Myh7,* which are upregulated in cardiomyopathy (Fig 7D–F). These findings indicate that prevention of mitochondrial iron accumulation through a pharmacological approach delays the development of cardiomyopathy in an animal model of mito-chondrial iron accumulation.

**Modulation of mitochondrial iron influences the formation of reactive oxygen species and mitochondrial complex activity after oxidative stress**

Although iron can catalyze ROS production through the Fenton reaction, it is not known whether iron accumulation can also affect mitochondrial respiratory chain complexes (which are the major sources of ROS production in mitochondria (Murphy, 2009; Schieber & Chandel, 2014)) and increase the cellular ROS derived from these structures. To determine whether changes in mitochondrial iron have any effect on mitochondrial complex I and complex III ROS production, we first modulated mitochondrial iron through overex-pression or downregulation of ABCB8 in H9c2 cells, followed by

measurement of ROS production at baseline. ROS production through complex I was measured in the presence of its substrates malate/pyruvate, while ROS production through complex III was assessed using succinate, the substrate for complex II which directly feeds complex III. Maximal ROS production was measured in the presence of rotenone (for complex I) and antimycin A (for complex III). ABCB8 overexpression was confirmed by Western blotting (Fig 8A). We did not observe any changes in baseline or maximal complex I and complex III ROS production with either downregula-tion (Fig 8B and C) or overexpression of ABCB8 (Fig 8D and E), suggesting that changes in baseline mitochondrial iron do not influ-ence the intrinsic ability of mitochondrial complexes to produce ROS.

Since iron can catalyze the formation of hydroxyl free radicals from hydrogen peroxide and cause cellular damage, we then studied whether modulation of mitochondrial iron influences ROS produc-tion and mitochondrial membrane potential *in vitro* after injury. Mitochondrial ROS, measured by mitoSox, was increased in H9c2 cells after hydrogen peroxide treatment (Fig 8F). The increase in mitochondrial ROS was significantly attenuated in cells with BPD pre-treatment, while DFO pre-treatment had no effects. Although this experiment does not directly measure the production of the

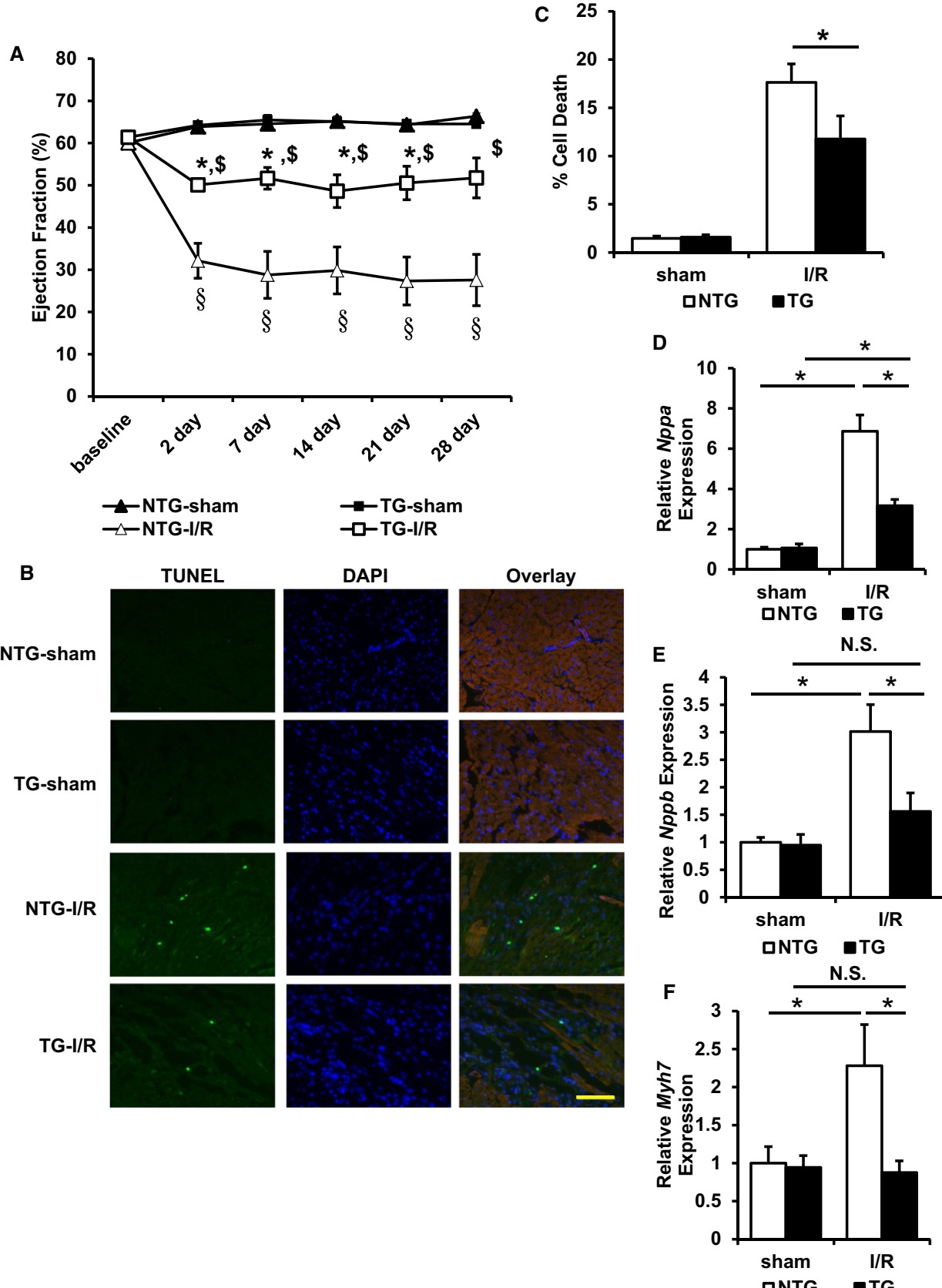

**Figure 4.**

◀

**Figure 4.  ABCB8 transgenic mice have less cell death and better cardiac function after I/R injury.**

A   Cardiac function in ABCB8 NTG and TG mice undergoing sham or I/R procedure. ANOVA followed by *post hoc* Tukey's test was performed for each time point. *$P$ < 0.05 compared with TG-sham at the same time point. $^{\$}P$ < 0.05 compared with NTG-I/R at the same time point. $^{\S}P$ < 0.05 compared with NTG-sham at the same time point. $N$ = 5 mice for NTG-sham and NTG-I/R and $N$ = 6 mice for TG-sham and TG-I/R. Exact *P*-values are included in Appendix Table S3.

B   Representative images of TUNEL staining for mice with the indicated genotype undergone the indicated procedure. Scale bar, 100 μm.

C   Quantification of apoptosis in ABCB8 TG and NTG mice with the indicated procedure. ANOVA followed by *post hoc* Tukey's test was performed. *$P$ = 0.035. $N$ = 4 mice for TG-I/R and $N$ = 6 mice for all other groups.

D   *Nppa* expression in mice subjected to sham or I/R procedure. ANOVA followed by *post hoc* Tukey's test was performed. *$P$ = 3.24E-4 NTG-sham vs. NTG-I/R. *$P$ = 0.002 TG-sham vs. TG-I/R. *$P$ = 8E-6 NTG-I/R vs. TG-I/R. $N$ = 5 mice for NTG-sham and NTG-I/R and $N$ = 6 mice for TG-sham and TG-I/R.

E   *Nppb* expression in mice subjected to sham or I/R procedure. ANOVA followed by *post hoc* Tukey's test was performed. *$P$ < 1E-8 NTG-sham vs. NTG-I/R. *$P$ = 0.032 NTG-I/R vs. TG-I/R. $N$ = 5 mice for NTG-sham and NTG-I/R and $N$ = 6 mice for TG-sham and TG-I/R.

F   *Myh7* expression in mice subjected to sham or I/R procedure. ANOVA followed by *post hoc* Tukey's test was performed. *$P$ = 0.04 NTG-sham vs. NTG-I/R. *$P$ = 0.034 NTG-I/R vs. TG-I/R. $N$ = 5 mice for NTG-sham and NTG-I/R and $N$ = 6 mice for TG-sham and TG-I/R.

Data information: All data are expressed as mean ± SEM. N.S., not significant.

hydroxyl free radicals, it provides a measure of overall mitochondrial ROS levels after $H_2O_2$ injury. Additionally, while chelator treatment did not have significant effects on mitochondrial membrane potential at baseline, BPD prevented the significant decrease of mitochondrial membrane potential after oxidative stress (Fig 8G). Thus, our findings are consistent with our hypothesis that lower mitochondrial iron at baseline is associated with less mitochondrial ROS and cellular damage after injury.

To investigate the effects of modulating mitochondrial iron levels on ROS production *in vivo* after injury, we measured lipid peroxidation products in chelator-treated ABCB8 KO mice and in ABCB8 transgenic mice that had undergone I/R. While ABCB8 KO mice had significantly higher levels of lipid peroxidation products, BPD treatment attenuated the increase (Fig 8H). Similarly, I/R injury in NTG mice resulted in a higher level of lipid peroxidation products, but the increase was attenuated with ABCB8 overexpression (Fig 8I). The results indicate that mitochondrial iron modulation has an effect on ROS production during cardiac injury.

Since increased mitochondrial ROS can inhibit mitochondrial TCA cycle enzymes and respiratory chain complexes, we measured the enzymatic activity of mitochondrial aconitase, complex I, II, and IV. Mitochondrial aconitase has a labile Fe/S cluster and is prone to oxidative damage (Vásquez-Vivar *et al*, 2000). Similarly, a decrease in the activity of complex I, II, and IV after oxidative damage has been described previously (Long *et al*, 2004; Moser *et al*, 2009; Wu *et al*, 2010). While ABCB8 overexpression did not cause changes in any of the enzyme activities, it prevented the decrease in mitochondrial aconitase, complex I, II, and IV activities after $H_2O_2$ challenge (Fig 9A–D). Similarly, while iron chelator treatment did not cause any difference in the activity of these enzymes, BPD pre-treatment protected the activities of mitochondrial aconitase, complex I, II, and IV after $H_2O_2$ treatment. On the other hand, DFO pre-treatment did not confer any protection (Fig 9E–H). These findings suggest that preventing ROS production due to mitochondrial iron accumulation during cardiac injury is one of the major mechanisms by which modulation of baseline mitochondrial iron levels is protective.

## Changes in mitochondrial iron are not associated with alterations of mitochondrial biogenesis or NOS expression

Mitochondrial dynamics have been shown to be associated with mitochondrial ROS production (Pletjushkina *et al*, 2006; Jheng *et al*, 2012), and the mitochondrial fission protein dynamin-related

protein 1 (Drp1) has been linked to apoptosis (Frank *et al*, 2001). To assess the role of mitochondrial dynamics in the observed protective effects of mitochondrial iron modulation, we measured the expression of genes involved in mitochondrial dynamics in H9c2 cells with ABCB8 overexpression or after treatment with iron chelators. No difference in the gene expression of mitochondrial fusion proteins (optic atrophy 1 (*Opa1*), mitofusin 1 (*Mfn1*), and mitofusin 2 (*Mfn2*)) and fission proteins (fission, mitochondrial 1 (*Fis1*), and *Drp1*) was observed in either groups, except for a decrease of *Fis1* seen in ABCB8 overexpressing H9c2 cells (Appendix Fig S7A and B). We also measured expression of genes associated with mitochondrial biogenesis and mitochondrial DNA content in these groups and found no difference (Appendix Fig S7C–F). Additionally, the expression of genes involved in mitochondrial biogenesis and mitochondrial dynamics was not altered by pharmacological or genetic modulation of mitochondrial iron in mice (Appendix Fig S8).

We also evaluated the expression of genes involved in mitochondrial biogenesis and mitochondrial dynamics in NTG and ABCB8 TG mice 2 day after I/R. mRNA levels of genes associated with mitochondrial biogenesis and mitochondrial dynamics were reduced to the same degree in NTG and ABCB8 TG mice in response to I/R (with the exception of nuclear respiratory factor 1, whose expression was not altered by I/R in either group) (Appendix Fig S9). Our observed changes of gene expression after I/R are consistent with published gene microarray datasets (GEO Accession number: GSE61592, GSE4105) (Roy *et al*, 2006). Taken together, our results indicate that modulation of mitochondrial iron has no effect on mitochondrial biogenesis or dynamics both at baseline and after I/R.

Nitric oxide synthase (NOS) has also been implicated in I/R injury (Davidson & Duchen, 2006). Therefore, we examined the expression of NOS genes both at baseline and after I/R in NTG and ABCB8 TG mice. Our results showed no difference in the expression of *Nos3* and *Nos2* (encoding eNOS and iNOS, respectively) between TG and NTG mice both at baseline and 2 days after sham or I/R operation (Appendix Fig S10A and B). Since NOS uncoupling secondary to the loss of its cofactor, $BH_4$, can result in ROS production (Verhaar *et al*, 2004), we also measured the expression of genes in the $BH_4$ synthesis pathway and found no difference between NTG and ABCB8 TG mice at baseline or 2 days after sham or I/R procedures (Appendix Fig S10C and D). Collectively, our data indicate that a decrease in mitochondrial iron has no effect on the expression of NOS proteins or their uncoupling. Therefore, the protective

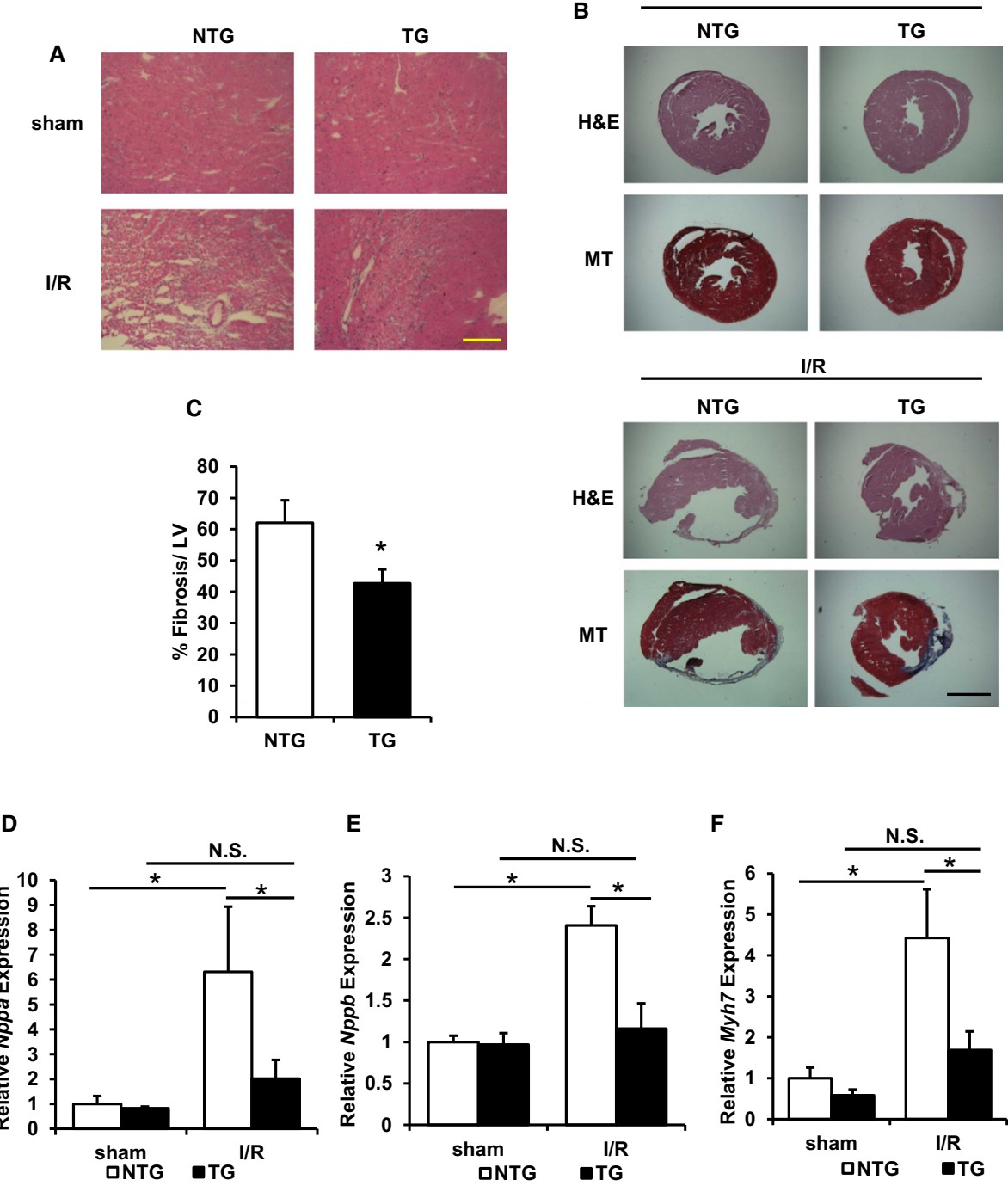

**Figure 5.  ABCB8 transgenic mice demonstrate reduced acute cellular injury and decreased left ventricular fibrosis and cardiac stress 28 days after injury.**

A   Representative H&E staining of peri-infarct area demonstrated reduced cellular injury in ABCB8 TG mice after I/R. Scale bar, 115 μm.

B   Representative H&E and Masson's trichrome (MT) staining in mouse heart 28 days after the indicated procedure. Scale bar, 1,100 μm.

C   Quantification of tissue fibrosis in ABCB8 NTG and TG mice subjected to I/R. *P = 0.047 with two-tailed unpaired t-test. Two to three sections from each mice were quantified; N = 4 mice for NTG and N = 6 mice for TG.

D   Nppa expression in mice subjected to sham or I/R procedure. ANOVA followed by post hoc Tukey's test was performed. *P = 0.006 NTG-sham vs. NTG-I/R. *P = 0.015 NTG-I/R vs. TG-I/R. N = 5 mice for NTG-sham and TG-I/R, N = 6 mice for TG-sham, and N = 4 mice for NTG-I/R.

E   Nppb expression in mice subjected to sham or I/R procedure. ANOVA followed by post hoc Tukey's test was performed. *P = 0.001 NTG-sham vs. NTG-I/R. *P = 0.004 NTG-I/R vs. TG-I/R. N = 5 mice for NTG-sham and TG-I/R, N = 6 mice for TG-sham, and N = 4 mice for NTG-I/R.

F   Myh7 expression in mice subjected to sham or I/R procedure. ANOVA followed by post hoc Tukey's test was performed. *P = 0.002 NTG-sham vs. NTG-I/R. *P = 0.011 NTG-I/R vs. TG-I/R. N = 5 mice for NTG-sham and TG-I/R, N = 6 mice for TG-sham, and N = 4 mice for NTG-I/R.

Data information: All data are expressed as mean ± SEM. N.S., not significant.

**Figure 6.  Pharmacological modulation of mitochondrial iron protects against cellular damage after I/R.**

A   Mitochondrial non-heme iron in wild-type mice treated with vehicle control or the indicated iron chelator for 7 days. ANOVA followed by *post hoc* Tukey's test was performed. *$P$ = 0.03. $N$ = 4 mice for PBS and DFO, and $N$ = 5 mice for BPD.

B   Cytosolic non-heme iron in wild-type mice treated with vehicle control or the indicated iron chelator for 7 days. ANOVA followed by *post hoc* Tukey's test was performed. *$P$ = 5E-4 PBS vs. DFO. *$P$ = 0.01 DFO vs. BPD. $P$ = 0.06 PBS vs. DFO. $N$ = 4 mice for PBS and DFO, and $N$ = 5 mice for BPD.

C   Nuclear non-heme iron in wild-type mice treated with vehicle control or the indicated iron chelator for 7 days. $N$ = 4 mice for PBS and DFO and $N$ = 5 mice for BPD.

D   Cardiac function in wild-type mice treated with vehicle control or the indicated iron chelator for 7 days. $N$ = 5 mice for each group.

E   Cardiac function of chelator-treated mice after I/R. ANOVA followed by *post hoc* Tukey's test was performed for each time point. *$P$ < 0.0001 compared to PBS-I/R group at the same time point. [#]$P$ < 0.0001 compared to PBS-sham group at the same time point. Exact $P$-values are included in Appendix Table S3. $N$ = 5 mice PBS-sham and BPD-I/R and $N$ = 6 mice for all other groups.

F   Representative hematoxylin and eosin (H&E) and Masson's trichrome (MT) staining of heart sections in mice with the indicated chelator treatment undergone sham or I/R. Scale bar, 1,100 μm. Bar graph represents the quantification of tissue fibrosis. ANOVA followed by *post hoc* Tukey's test was performed. *$P$ = 0.048. Two to three sections from each mice were quantified, $N$ = 4 mice for PBS, $N$ = 5 mice for DFO, and $N$ = 6 mice for BPD.

G   *Nppa* expression in mice subjected to sham or I/R procedure. ANOVA followed by *post hoc* Tukey's test was performed. *$P$ = 0.002 PBS-sham vs. PBS-I/R. *$P$ = 0.02 DFO-sham vs. DFO-I/R. *$P$ = 0.007 PBS-I/R vs. BPD-I/R. $N$ = 6 mice for PBS-sham and DFO-sham and $N$ = 4 mice for all other groups.

H   *Nppb* expression in mice subjected to sham or I/R procedure. ANOVA followed by *post hoc* Tukey's test was performed. *$P$ = 0.0007 PBS-sham vs. PBS-I/R. *$P$ = 0.014 DFO-sham vs. DFO-I/R. *$P$ = 0.003 PBS-I/R vs. BPD-I/R. $N$ = 6 mice for PBS-sham and DFO-sham, $N$ = 4 mice for all other groups.

I   *Myh7* expression in mice subjected to sham or I/R procedure. ANOVA followed by *post hoc* Tukey's test was performed. *$P$ = 0.0001 PBS-sham vs. PBS-I/R. *$P$ = 0.038 DFO-sham vs. DFO-I/R. *$P$ = 0.006 PBS-I/R vs. BPD-I/R. $N$ = 6 mice for PBS-sham and DFO-sham and $N$ = 4 mice for all other groups.

Data information: All data are expressed as mean ± SEM. N.S., not significant.

effects of mitochondrial iron modulation cannot be attributed to a NOS-mediated mechanism.

# Discussion

The association between changes in iron and ischemic injury has been demonstrated in various organ systems (Coudray *et al*, 1994; Zhao *et al*, 1997; Comporti *et al*, 2002; Kaushal & Shah, 2014). These observations gave rise to the hypothesis that modulation of iron is protective against ischemic damage. However, the effects of iron chelation are very inconsistent among different studies. Importantly, none of the previous studies had focused on the role of iron in specific intracellular locations. Based on our observation that mitochondrial iron is significantly increased in mice after I/R, in cells treated with $H_2O_2$, and in human patients with ISCM, we hypothesized that a specific reduction in mitochondrial iron would offer protection against cardiac injury. We further hypothesized that the discrepancies observed between the efficacies of iron chelators in earlier studies could be due to intrinsic differences in their ability to penetrate the mitochondria. Here, we show that a decrease in mitochondrial iron at baseline *in vivo* using either genetic or pharmacological approaches is protective against I/R damage. Additionally, we demonstrate that preventing mitochondrial iron accumulation can be a viable therapeutic approach against the deterioration of cardiac function in a mouse model of genetic mitochondrial iron overload in the heart. The protective effects seen in both models can be attributed to a reduction in ROS production during injury. Thus, our studies highlight the importance of mitochondrial iron in I/R damage.

Our findings are consistent with a recently published randomized clinical trial showing that chelation therapy using EDTA in patients with myocardial infarction reduced adverse cardiovascular outcomes (Lamas *et al*, 2013), as well as other studies suggesting therapeutic benefits of iron chelation in heart disease (Badylak *et al*, 1987; Ramesh Reddy *et al*, 1989; Lesnefsky *et al*, 1990b; Kobayashi *et al*, 1991; Williams *et al*, 1991; Chopra *et al*, 1992; Nicholson *et al*, 1997). However, a controversy in this field pertains to the use

of iron chelation or iron supplementation as a therapy for patients with heart failure. Other recent clinical trials suggest that patients with heart failure and iron deficiency may benefit from iron supplementation (Anker *et al*, 2009; Ponikowski *et al*, 2014), and there are currently attempts to initiate a phase III clinical study to assess the effects of intravenous iron infusion in heart failure. The discrepancy between this and our studies can be explained by the discordance between systemic iron status and cellular iron status. While cardiac mitochondrial iron overload occurs in heart failure patients (Khechaduri *et al*, 2013), they may be simultaneously iron deficient at the systemic level. Reversal of iron deficiency through intravenous iron may correct many patients' symptoms due to its effect on hematopoiesis and account for the improvement in patients' functional capacities. However, changes in systemic iron status are physiologically distinct from disturbances in subcellular iron homeostasis, and mitochondrial iron overload in the heart would persist in these patients. Based on our data, decreasing iron in this compartment would have beneficial effects on cardiac disease progression.

Mitochondria contain 10–20 μM of labile iron, but the exact concentration can vary from organ to organ (Petrat *et al*, 2002a; Rauen *et al*, 2007). While the level of "free" iron (not bound to any ligands) in the mitochondria is not readily determined, it is expected to be low given the tight regulation of cellular iron flux and storage. However, ROS is known to damage iron-containing molecules, especially Fe/S clusters, and free-up incorporated iron (Flint *et al*, 1993; Brazzolotto *et al*, 1999; Jang & Imlay, 2007; Cantu *et al*, 2009). Therefore, in I/R, in which ROS levels are increased, the amount of free iron in the mitochondria may be higher than basal conditions. Increased labile iron may also disrupt mitochondrial iron homeostasis, resulting in the increased mitochondrial iron observed in the acute phase of I/R injury in our *in vivo* studies. Furthermore, increased lysosomal delivery of iron to mitochondria during ischemia can also lead to increased mitochondrial iron levels (Zhang & Lemasters, 2013). It is possible that all of these mechanisms contribute to the increase in mitochondrial iron in mouse hearts after I/R. The increased iron level in turn further augments ROS production, which can inhibit mitochondrial aconitase, complex I,

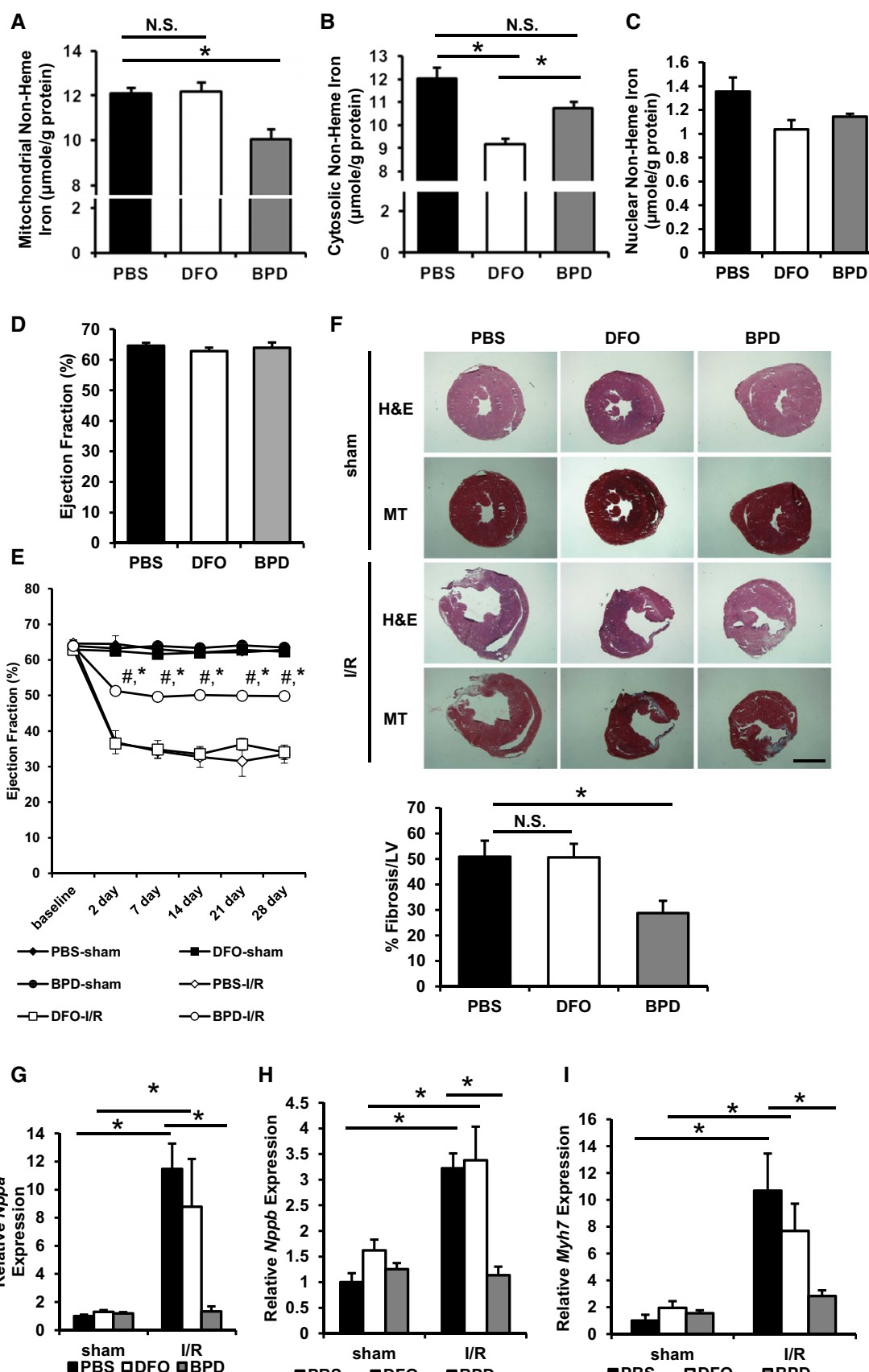

**Figure 6.**

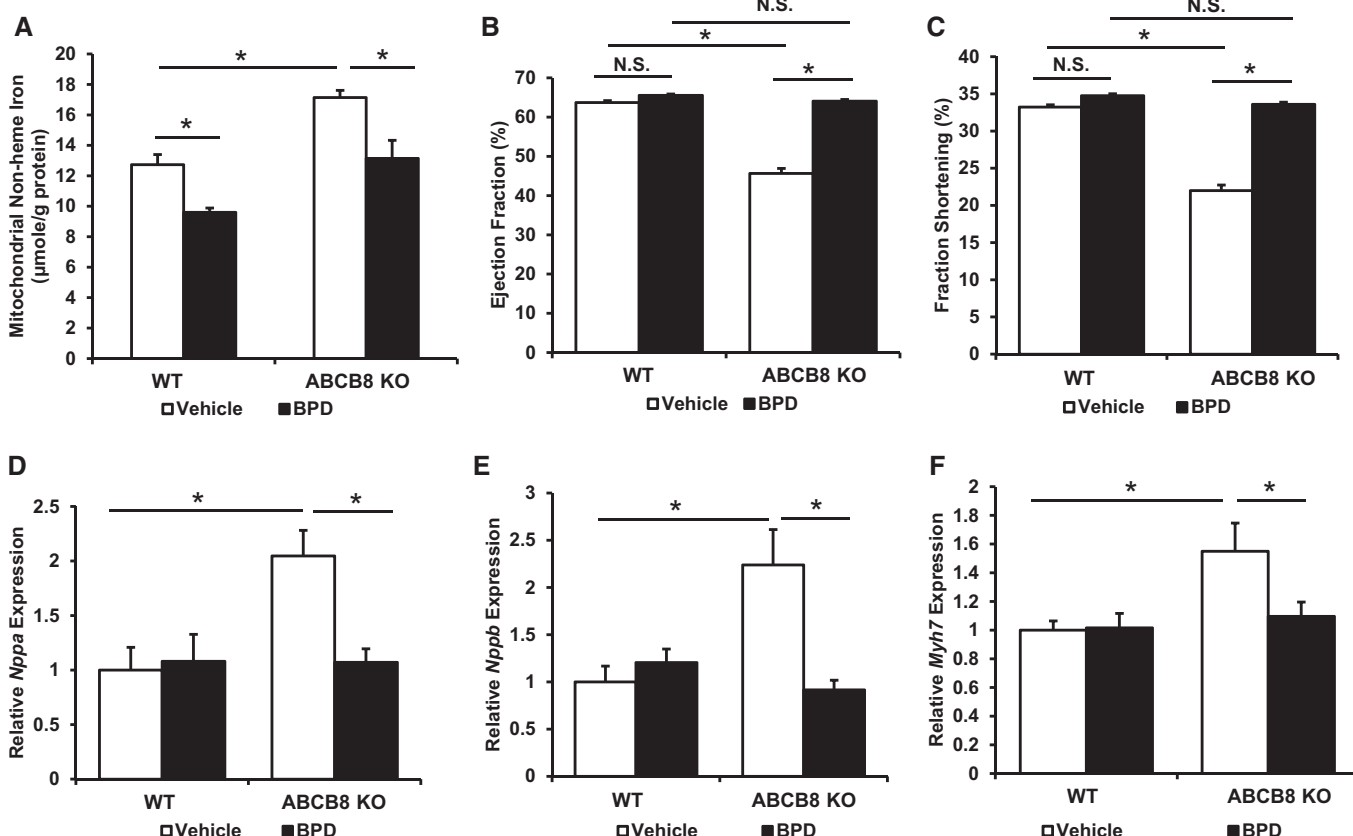

**Figure 7.  A decrease in mitochondrial iron at baseline protects against the development of cardiomyopathy in inducible cardiac-specific ABCB8 knockout mice.**

A   Cardiac mitochondrial iron in ABCB8 KO and WT mice with the indicated treatment harvested 4 weeks after the tamoxifen treatment. ANOVA followed by *post hoc* Tukey's test was performed. *P = 0.02 WT vehicle vs. WT BPD. *P = 0.001 WT vehicle vs. ABCB8 KO vehicle. *P = 0.004 ABCB8 KO vehicle vs. ABCB8 KO BPD. N = 8 for WT vehicle and N = 7 for all other groups.

B   Ejection fraction in ABCB8 KO and WT mice with the indicated treatment 4 weeks after gene knockout. ANOVA followed by *post hoc* Tukey's test was performed. *P < 1E-8 WT vehicle vs. ABCB8 KO vehicle. *P < 1E-8 ABCB8 KO vehicle vs. ABCB8 KO BPD. N = 8 for WT vehicle and N = 7 for all other groups.

C   Fractional shortening in ABCB8 KO and WT mice with the indicated treatment 4 weeks after gene knockout. ANOVA followed by *post hoc* Tukey's test was performed. *P < 1E-8 WT vehicle vs. ABCB8 KO vehicle. *P < 1E-8 ABCB8 KO vehicle vs. ABCB8 KO BPD. N = 8 for WT vehicle and N = 7 for all other groups.

D   *Nppa* expression in mice subjected to sham or I/R procedure. ANOVA followed by *post hoc* Tukey's test was performed. *P = 0.017 WT vehicle vs. ABCB8 KO vehicle. *P = 0.01 ABCB8 KO vehicle vs. ABCB8 KO BPD. N = 8 for WT vehicle and N = 7 for all other groups.

E   *Nppb* expression in mice subjected to sham or I/R procedure. ANOVA followed by *post hoc* Tukey's test was performed. *P = 0.001 WT vehicle vs. ABCB8 KO vehicle. *P = 0.0003 ABCB8 KO vehicle vs. ABCB8 KO BPD. N = 8 for WT vehicle and N = 7 for all other groups.

F   *Myh7* expression in mice subjected to sham or I/R procedure. ANOVA followed by *post hoc* Tukey's test was performed. *P = 0.011 WT vehicle vs. ABCB8 KO vehicle. *P = 0.045 ABCB8 KO vehicle vs. ABCB8 KO BPD. N = 8 for WT vehicle and N = 7 for all other groups.

Data information: All data are expressed as mean ± SEM. N.S., not significant.

and complex II activities (as observed in ours and other studies (Chen & Zweier, 2014)), and can cause cytochrome C release secondary to AMPK activation (Dixon & Stockwell, 2014). Therefore, a decrease in baseline mitochondrial iron could lead to less "free" and total iron during oxidative stress, thereby reducing iron-catalyzed ROS production and cell death.

Earlier studies have suggested that DFO might protect radiation-mediated or low-dose $H_2O_2$-induced cell death through a lysosomal iron-dependent mechanism (Yu *et al*, 2003; Persson *et al*, 2005). Although lysosomal iron may contribute to the I/R injury, damage to mitochondrial respiratory chain activity, mitochondrial membrane lipids and mitochondrial DNA from mitochondria-derived ROS are likely to be more important during tissue I/R. As DFO predominantly exerts its effect through iron binding in the

extracellular space and endosome (Lloyd *et al*, 1991; Doulias *et al*, 2003), the inability to modulate mitochondrial iron can explain the lack of efficacy of DFO in our *in vivo* experiments, which is also consistent with some other large animal studies as well as a recent clinical trial (Ramesh Reddy *et al*, 1989; Lesnefsky *et al*, 1990b; Kobayashi *et al*, 1991; Chopra *et al*, 1992; Chan *et al*, 2012). In contrast, the ability of BPD to penetrate into subcellular compartments (Demougeot *et al*, 2004) may explain its protective effects in our *in vitro* and *in vivo* models. While iron chelators also have systemic effects and can modulate mitochondrial iron in other organs in addition to the heart, our genetic model specifically lowers mitochondrial iron in the cardiomyocyte. Therefore, the results from these two models argue that cardiac mitochondrial iron plays a causative role in I/R damage. These findings, combined with our

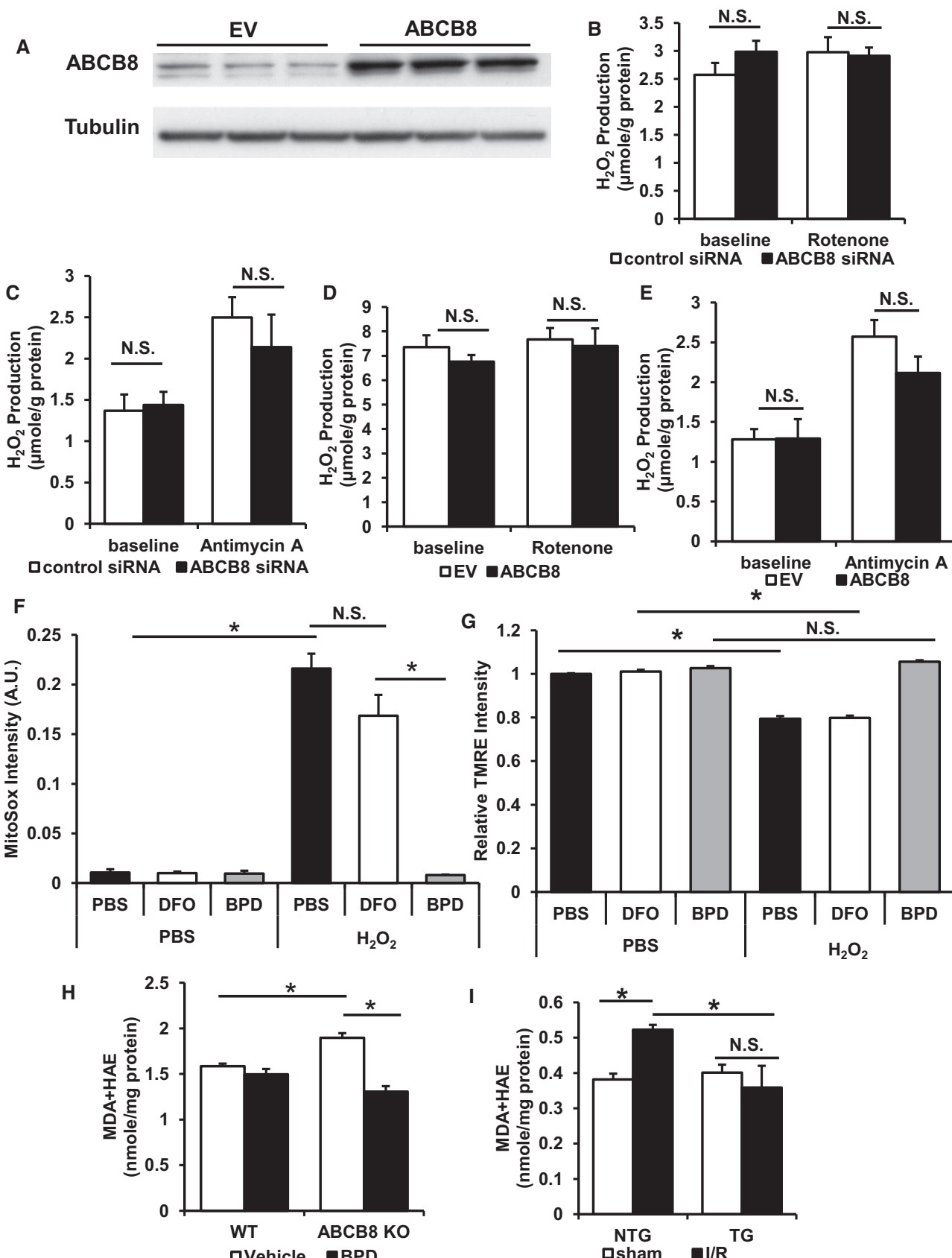

**Figure 8.**

◀

**Figure 8.   Modulation of mitochondrial iron influences the production of mitochondrial ROS.**

A   Representative Western blot demonstrating ABCB8 overexpression in H9c2 cells. EV, empty vector. $N = 3$ independent samples for each group.
B   Complex I ROS production in mitochondria with ABCB8 downregulation with or without rotenone. ANOVA followed by *post hoc* Tukey's test was performed. $N = 6$ independent samples for each group.
C   Complex III ROS production in mitochondria with ABCB8 downregulation with or without antimycin A. ANOVA followed by *post hoc* Tukey's test was performed. $N = 6$ independent samples for each group.
D   Complex I ROS production in mitochondria with ABCB8 overexpression with or without rotenone. EV, empty vector. ANOVA followed by *post hoc* Tukey's test was performed. $N = 9$ independent samples for EV baseline and $N = 8$ independent samples for the other groups.
E   Complex III ROS production in mitochondria with ABCB8 overexpression with or without antimycin A. EV, empty vector. ANOVA followed by *post hoc* Tukey's test was performed. $N = 9$ independent samples for EV baseline and $N = 8$ independent samples for the other groups.
F   Mitochondrial ROS in H9c2 cells with various treatments with or without oxidative stress. $N = 6$ independent samples for each group. ANOVA followed by *post hoc* Tukey's test was performed. *$P = $ 2E-8 PBS-PBS vs. PBS-$H_2O_2$. *$P = $ 2E-8 DFO-$H_2O_2$ vs. BPD-$H_2O_2$.
G   Mitochondrial membrane potential as measured by TMRE intensity in H9c2 cells treated with the indicated iron chelator with or without oxidative stress. $N = 6$ independent samples for each group. ANOVA followed by *post hoc* Tukey's test was performed. *$P = $ 1E-8 PBS-PBS vs. PBS-$H_2O_2$. *$P = $ 1.8E-8 DFO-PBS vs. DFO-$H_2O_2$.
H   The levels of lipid peroxidation products in ABCB8 KO mice with or without chelator treatment. $N = 4$ mice for WT vehicle, $N = 6$ mice for ABCB8 KO vehicle, and $N = 5$ mice for all other groups. ANOVA followed by *post hoc* Tukey's test was performed. *$P = 0.004$ WT vehicle vs. ABCB8 KO vehicle. *$P = $ 1E-6 ABCB8 KO vehicle vs. ABCB8 KO BPD.
I   Lipid peroxidation products in ABCB8 TG and NTG mice 2 days after I/R. $N = 6$ mice for NTG-sham and $N = 4$ mice for the other groups. ANOVA followed by *post hoc* Tukey's test was performed. *$P = 0.024$ NTG-sham vs. NTG-I/R. *$P = 0.016$ NTG-I/R vs. TG-I/R.

Data information: All data are expressed as mean $\pm$ SEM. N.S., not significant.
Source data are available online for this figure.

observation of increased mitochondrial iron in I/R, underscore the significance of targeting mitochondrial iron in developing future therapies.

Our transgenic model of cardiac-specific ABCB8 overexpression displayed lower levels of mitochondrial iron while cytosolic iron remained similar. The lower amount of mitochondrial iron can be due to increased iron export, as seen in other studies (Ichikawa *et al*, 2012). Since the majority of intracellular iron is stored in ferritin molecules in the cytoplasm, mitochondrial iron only represents a small fraction of the total cellular iron. This is supported by experiments in which cells are incubated with a radioactive iron chaser and the majority of radioactivity concentrates in the cytoplasm. Therefore, the approximately 20% decrease in mitochondrial iron seen in our transgenic model might not significantly affect the total cellular iron level.

Although targeting whole-cell ROS using antioxidants did not show benefits in different disease settings, including cardiovascular disease and cancer (Jha *et al*, 1995; Steinhubl, 2008; Fortmann *et al*, 2013), recent animal studies and human trials using mitochondria-targeted antioxidants have offered promising results (Adlam *et al*, 2005; Neuzil *et al*, 2007; Xu *et al*, 2008; Gane *et al*, 2010; Dai *et al*, 2011). Pre-clinical studies also highlighted the involvement of complex III-mediated ROS production during tissue ischemia (Lesnefsky & Hoppel, 2003). These findings indicate that targeting antioxidants to the proper site of ROS production is critical. While our studies suggest that modulation of mitochondrial iron does not have an effect on baseline ROS production, the excess free iron during I/R can convert the ROS from mitochondrial complexes and generate more damage, which would explain the difference in lipid peroxidation products in our *in vivo* system. Our studies emphasize the importance of targeting mitochondrial iron (which ultimately results in a reduction in mitochondrial ROS) during I/R injury, and the design of future therapies should take into consideration the subcellular specificity of the intervention.

In summary, we demonstrated that mitochondrial iron is a key player in ischemic damage to the heart. Genetic and pharmacological approaches lowering mitochondrial iron at baseline led to reduced cardiac damage from I/R. We also showed that mitochondrial iron plays a causative role in the development of cardiomyopathy in a genetic model of cardiac mitochondrial iron accumulation. Lastly, the protective effect of modulating baseline mitochondrial iron is at least partially through a reduction in mitochondrial ROS production.

# Materials and Methods

### Human heart failure tissue samples

Non-failing and ischemic cardiomyopathy cardiac tissue samples were obtained from the Human Heart Tissue Collection at the Cleveland Clinic. Informed consent was obtained from all the transplant patients and from the families of the organ donors before tissue collection. Protocols for tissue procurement were approved by the Institutional Review Board of the Cleveland Clinic (Cleveland, Ohio, USA), which is AAHRPP accredited. The experiments conformed to the principles set out in the WMA Declaration of Helsinki and the Department of Health and Human Services Belmont Report.

### Mouse model

Cardiomyocyte-specific ABCB8 transgenic mice (C57/BL6 background) were generated as described previously (Ichikawa *et al*, 2014) and backcrossed for at least eight generations. Mice with both alleles of ABCB8 floxed (ABCB8$^{f/f}$, C57/BL6 background) were crossed to α-MHC-MER-Cre-MER mice (C57/BL6 background), and the first generation is backcrossed with ABCB8$^{f/f}$ mice to generate ABCB8$^{f/f}$ mice with or without α-MHC-MER-Cre-MER transgene. Tamoxifen-induced Cre translocation and deletion of ABCB8 in the heart was achieved using a previously described protocol at 8 weeks of age (Ichikawa *et al*, 2012). Both male and female ABCB8$^{f/f}$ mice with or without α-MHC-MER-Cre-MER transgene were used in the studies. All mice were housed in the barrier facility at Northwestern University with 12-h light and 12-h dark

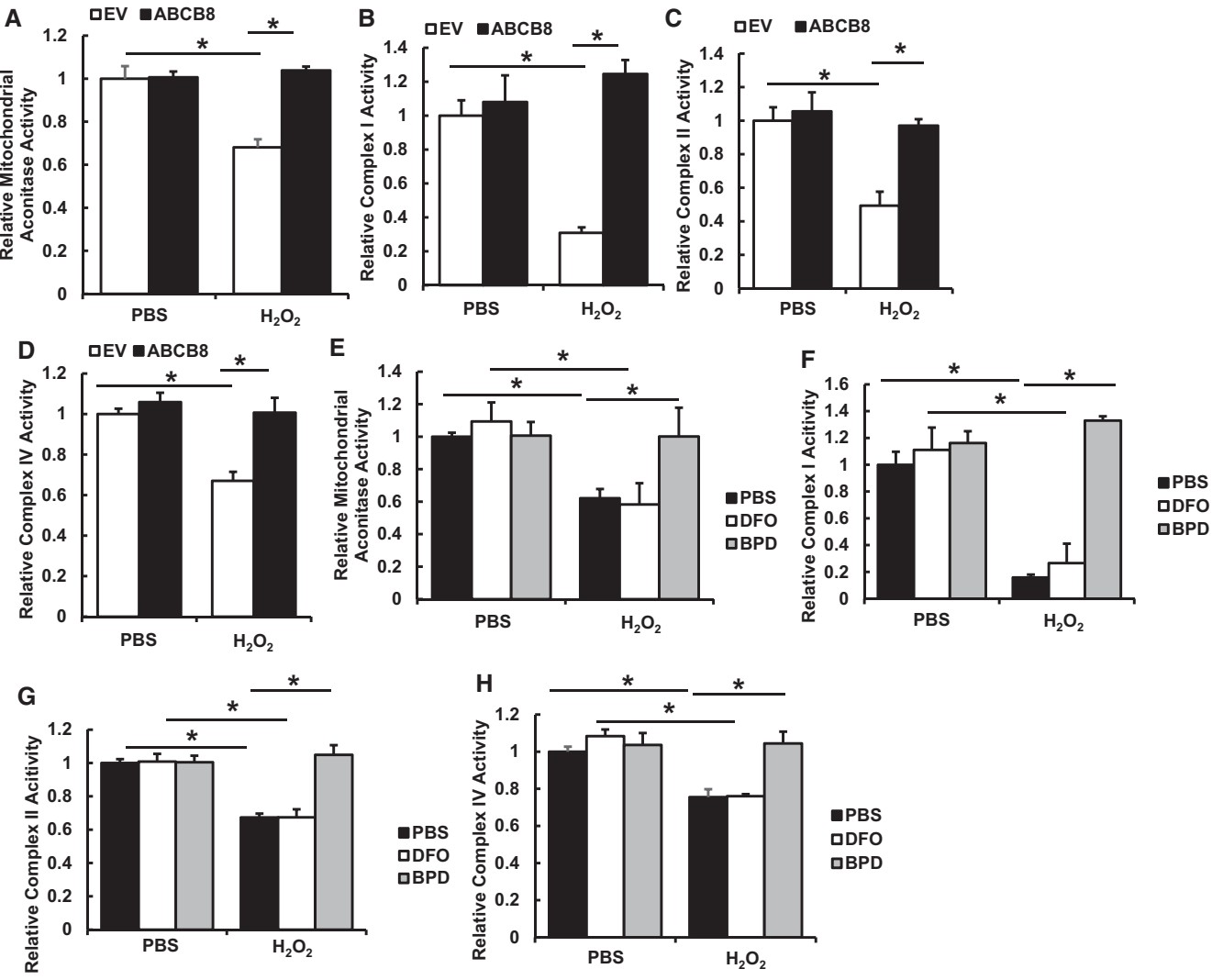

**Figure 9. Modulation of mitochondrial iron preserves the activity of mitochondrial complexes after oxidative stress.**

A  Mitochondrial aconitase activity in H9c2 cells with or without ABCB8 overexpression and treated with or without $H_2O_2$. $N = 4$ independent samples for each group. ANOVA followed by *post hoc* Tukey's test was performed. *$P = 0.0004$ EV-PBS vs. EV-$H_2O_2$. *$P = 0.0001$ EV-$H_2O_2$ vs. ABCB8-$H_2O_2$.

B  Complex I activity in H9c2 cells with or without ABCB8 overexpression and treated with or without $H_2O_2$. $N = 6$ independent samples for each group. ANOVA followed by *post hoc* Tukey's test was performed. *$P = 0.0005$ EV-PBS vs. EV-$H_2O_2$. *$P = 0.00001$ EV-$H_2O_2$ vs. ABCB8-$H_2O_2$.

C  Complex II activity in H9c2 cells with or without ABCB8 overexpression and treated with or without $H_2O_2$. $N = 6$ independent samples for each group. ANOVA followed by *post hoc* Tukey's test was performed. *$P = 0.001$ EV-PBS vs. EV-$H_2O_2$. *$P = 0.001$ EV-$H_2O_2$ vs. ABCB8-$H_2O_2$.

D  Complex IV activity in H9c2 cells with or without ABCB8 overexpression and treated with or without $H_2O_2$. $N = 6$ independent samples for each group. ANOVA followed by *post hoc* Tukey's test was performed. *$P = 0.0008$ EV-PBS vs. EV-$H_2O_2$. *$P = 0.0006$ EV-$H_2O_2$ vs. ABCB8-$H_2O_2$.

E  Mitochondrial aconitase activity in H9c2 cells treated with the indicated iron chelators with or without $H_2O_2$. $N = 4$ independent samples in each group. ANOVA followed by *post hoc* Tukey's test was performed. *$P = 0.036$ PBS-PBS vs. PBS-$H_2O_2$. *$P = 0.0047$ DFO-PBS vs. DFO-$H_2O_2$. *$P = 0.035$ PBS- $H_2O_2$ vs. BPD-$H_2O_2$.

F  Complex I activity in H9c2 cells treated with the indicated iron chelators with or without $H_2O_2$. $N = 6$ independent samples for each group. ANOVA followed by *post hoc* Tukey's test was performed. *$P = 0.0004$ PBS-PBS vs. PBS-$H_2O_2$. *$P = 0.0003$ DFO-PBS vs. DFO-$H_2O_2$. *$P < 0.0001$ PBS- $H_2O_2$ vs. BPD-$H_2O_2$.

G  Complex II activity in H9c2 cells treated with the indicated iron chelators with or without $H_2O_2$. $N = 6$ independent samples for each group. ANOVA followed by *post hoc* Tukey's test was performed. *$P = 0.0001$ PBS-PBS vs. PBS-$H_2O_2$. *$P < 0.0001$ DFO-PBS vs. DFO-$H_2O_2$. *$P < 0.0001$ PBS- $H_2O_2$ vs. BPD-$H_2O_2$.

H  Complex IV activity in H9c2 cells treated with the indicated iron chelators with or without $H_2O_2$. $N = 6$ independent samples for each group. ANOVA followed by *post hoc* Tukey's test was performed. *$P = 0.02$ PBS-PBS vs. PBS-$H_2O_2$. *$P = 0.0009$ DFO-PBS vs. DFO-$H_2O_2$. *$P = 0.006$ PBS- $H_2O_2$ vs. BPD-$H_2O_2$.

Data information: All data are expressed as mean $\pm$ SEM.

cycle and received normal chow and water *ad lib* unless otherwise indicated. All animal studies were approved by the Institutional Animal Care and Use Committee at Northwestern University and were performed in accordance with guidelines from the National Institutes of Health.

**Iron chelator treatment**

For I/R studies, wild-type mice littermates (C57/BL6) were randomized to receive 80 mg/kg BPD (Sigma) in normal saline solution daily or 50 mg/kg DFO (Sigma) in normal saline every other day or

vehicle daily via intraperitoneal injection. The dose of BPD was chosen based on its ability to modulate cardiac mitochondrial iron, and the dose of DFO was chosen according to previously published studies (Ichikawa et al, 2014). For ABCB8 KO and corresponding wild-type littermates, mice were randomized to receive BPD or vehicle daily via intraperitoneal injection. BPD was prepared as a 400 mg/ml stock solution in ethanol and diluted 1:100 in normal saline to working concentration. DFO was prepared as a 250 mg/ml stock in water and diluted 1:100 in normal saline to working concentration.

### Ischemia–reperfusion

For surgery, male mice at age 10–14 weeks were used. Age-matched littermates were used for each operation. Wild-type mice (C57/BL6) used in the chelators treatment studies received chelators treatment for 1 week prior to the operation. Mice with the same genotype or treatment were randomized to receive either I/R or sham operation. The surgeon was blinded to the genotype or the treatment of mice. The procedure was performed as described previously (Wu et al, 2011).

### Echocardiography

Parasternal short- and long-axis views of the heart were obtained using a Vevo 770 high-resolution imaging system with a 30 MHz scan head. 2D and M-mode images were obtained and analyzed. Ejection fraction was calculated from M-mode image using Teichholtz equation, and fractional shortening was directly calculated from end-systolic and end-diastolic chamber size from M-mode images.

### Histochemical analysis

At the time of tissue harvest, heart was excised and rinsed in phosphate-buffered saline to remove excess blood on tissue and in ventricles. Hearts were then submerged into OTC compound and frozen in liquid nitrogen. For hearts with I/R injury, sections were collected 500 μm below suture line to capture the injured region. Sections were stained with hematoxylin and eosin for evaluation of general cardiac morphology and tissue organization. Masson's trichrome staining was used to visualize cardiac fibrosis. Fibrosis was quantified from low power microscope images by dividing the arc length of fibrotic scar to the circumference of left ventricle.

For confocal microscopy, frozen sections were fixed in cold acetone. Non-specific antigen binding was blocked by incubating sections with 5% donkey serum prior to incubating sections with antibody against ABCB8 (1:100) (Ardehali et al, 2005) and COX4 (1:100, ab33985, Abcam) in 4°C overnight. Species-specific secondary antibody with different fluorophore (1:200, Jackson Immunochem) was used to visualize the antigen, and nucleus were counterstained with TO-PRO-3 stain (Life Technologies) according to manufacturer's instructions. Images were acquired on a Zeiss LSM510 confocal microscope.

### Cell culture

H9c2 cardiomyoblasts were purchased from ATCC and cultured in DMEM (ATCC) supplemented with 10% FBS (Thermo) and penicillin/streptomycin (Cellgro). HEK293T cells were purchased from ATCC and cultured in MEM (Cellgro) supplemented with 10% FBS, sodium pyruvate (Cellgro), and penicillin/streptomycin. The cells were used within 10 cultures after thawing for the experiments in this paper. Unless otherwise specified, cells were treated with 200 μM of DFO or 100 μM of BPD for 2 h when indicated prior to addition of 600 μM of $H_2O_2$ when indicated for 6 h.

### RNA isolation and qRT–PCR

RNA was isolated using RNA-STAT60 (Tel-Test) according to manufacturer's instructions and subjected to DNAse I (Ambion) digestion to remove residual DNA. Purified RNA was then reverse transcribed with random hexamer and oligo-dT(16) (Applied Biosystems) and amplified on a 7500 Fast Real-Time PCR system using Fast SYBR Green PCR Master Mix (Applied Biosystems). The sequences for primers are included in Appendix Table S2. mRNA levels were calculated based on the difference of threshold $C_t$ values in target gene and average $C_t$ values of 18s, Actb, B2m, and Hprt in the same sample.

### Isolation of mitochondria and nuclei

Mitochondria from tissues and cells were isolated via differential centrifugation using the Mitochondria Isolation Kit for Tissue and Mitochondria Isolation Kit for Mammalian Cells (Pierce), respectively. Nuclei were isolated using NE-PER Nuclear and Cytoplasmic Extraction Reagents (Pierce).

### Measurement of iron

Tissue iron was measured colorimetrically by the formation of a complex with bathophenanthroline disulfonate or ferene-S as described previously (Ichikawa et al, 2012; Khechaduri et al, 2013). For qualitative measurement of mitochondrial labile iron, cells were stained with rhodamine B-[(1,10-phenanthroline-5-yl)-aminocarbonyl]benzyl ester (RPA) (Squarix) and mitoTracker Green (Invitrogen). Images were obtained on a Zeiss AxioObserver.Z1 microscope. Quantitative measurement of labile iron was done according to previously described methods (Cabantchik et al, 1996; Petrat et al, 2002b). Briefly, cells were stained with RPA and mitoTracker Green for mitochondrial labile iron measurement and calcein (Invitrogen) and Hoechst 33342 (Invitrogen) for cytosolic labile iron measurement. Fluorescence intensity was measured using Gemini XS plate reader (Molecular Device) with following excitation emission setting: ex/em 490 nm/520 nm for mitoTracker Green and calcein, 564 nm/601 nm for RPA, and 350 nm/461 nm for Hoechst 33342. After initial measurement, 2 mM of PIH (Abcam) was added to each well and fluorescence signal was measured again. The difference of fluorescence intensity before and after PIH addition was used to calculate the concentration of iron-binding calcein or RPA based on a standard curve generated using fixed concentration of calcein or RPA. To account for difference in cell number due to plating or treatment with iron chelator and/or $H_2O_2$, the concentration of iron-binding RPA and calcein was normalized to mitoTracker Green and Hoechst 33342 signal of the same well, respectively.

For nuclear and cytosolic iron, cells were loaded with 280 nM of $^{55}$Fe-NTA. Excess $^{55}$Fe-NTA was washed off with cold PBS

containing 200 μM DFO prior to chelator treatment. The nuclear and cytosolic fractions were isolated using NE-PER Nuclear and Cytoplasmic Extraction Reagents (Pierce), and the radioactivity was quantified with liquid scintillation.

### Measurement of lipid peroxidation products

MDA and HAE in tissue samples were analyzed using the LPO microplate-based assay kit (Oxford Biochemical Research) according to manufacturer's instructions.

### Cell death studies

For *in vitro* studies, cells were resuspended in 1× Annexin V buffer (BD) and then stained with Alexa Fluor 350 conjugated Annexin V (Life Technology) and propidium iodide (Sigma) according to manufacturer's instructions and analyzed on a LSRII flow cytometer (BD).

For measuring apoptosis in tissue sections, sections were stained using an *in situ* cell death detection kit (Roche) according to manufacturer's instructions and then counterstained with DAPI (Sigma) and Cy5-Phalloidin (Molecular Probes). Images were obtained on a Zeiss AxioObserver.Z1 microscope.

### Overexpression and downregulation of ABCB8

For knockdown studies, pooled ABCB8 siRNA or control siRNA (Qiagen) was transfected into H9c2 cells using Dharmafect I reagent (GE Healthcare) for 72 h according to manufacturer's protocol. The effectiveness of this siRNA was verified before (Ichikawa *et al*, 2012). ABCB8 overexpression was achieved by cloning the ABCB8 coding sequence into the pHIV-eGFP vector (Addgene). Sequences were verified through direct sequencing. Virus was generated by transfecting HEK293T cells with packaging plasmid and viral construct using calcium phosphate transfection. Equal titers of ABCB8 or empty vector virus were used to infect H9c2 cells for 72 h. Overexpression was verified with Western blotting.

### Measurement of mitochondrial ROS production and mitochondrial ROS levels

Mitochondria were isolated as described before and resuspended in buffer containing 220 mM mannitol, 75 mM sucrose, 20 mM HEPEs (pH 7.4), 0.5 mM EDTA, 0.1 mM ATP, 0.5 mM magnesium acetate. Mitochondria were fueled with 10 mM of succinate or 3 mM sodium pyruvate and 3 mM sodium malate. ROS production was measured by Amplex Ultra Red (Life Technologies) according to manufacturer's instructions in the presence of 2 U/ml HRP and 200 U/ml SOD. Rotenone (complex I inhibitor, Sigma) and antimycin A (complex III inhibitor, Sigma) were added when indicated.

Mitochondrial ROS levels in intact cells were quantified using MitoSox Red (Life Technologies). Briefly, cells were loaded with MitoSox to stain for mitochondrial ROS and Hoechst 33342 for nuclei counter stain. Images were obtained on a Zeiss AxioObserver.Z1 microscope and analyzed with ImageJ software (NIH). MitoSox signal in the nuclei was subtracted to exclude localization of the dye to the nuclei.

### Western blotting

Whole-cell lysate or subcellular fractions were loaded onto 4–12% Bis-Tris acrylamide gel (Life Technology) and transferred to nitrocellulose membrane (GE Life Science). Membrane was incubated with primary antibody against ABCB8 (1:3,000) (Ardehali *et al*, 2005), tubulin (1:5,000, ab4074, Abcam), GAPDH (1:3,000, ab9485, Abcam), SDH70 kDa (1:5,000, 459200, Life Technologies), or lamin A/C (Cell Signaling) overnight in TBS with 0.05% Tween-20 and 5% milk. The membrane was then hybridized with horseradish peroxidase conjugated secondary antibody against rabbit or mouse (1:3,000, Jackson Immunochem), and the signal was visualized using Supersignal West Pico Substrate (Life Technologies).

### Mitochondrial enzyme and complex activities measurement

Mitochondria were isolated from cells as described above. Mitochondrial complex I, II, and IV activities were measured as described previously (Spinazzi *et al*, 2012). Aconitase activity was measured using Aconitase Activity Microplate Assay Kit (Abcam). Citrate synthase activity was measured from the same sample using Citrate Synthase Activity Kit (BioVision) and was used to normalize the mitochondrial complex activity.

### Mitochondrial DNA content measurement

Genomic and mitochondrial DNA from H9c2 cells were isolated using GeneJet DNA Isolation Kit (Thermo Scientific) according to manufacturer's instruction. The isolated DNA were diluted 1:10 and used as template for amplifying regions of ATP6 (mitochondrial DNA) and 18s (nuclear DNA) sequences. PCR was carried out on a 7500 Fast Real-Time PCR system using Fast SYBR Green PCR Master Mix (Applied Biosystems). The abundance of mitochondrial DNA was calculated based on difference of threshold $C_t$ values between ATP6 and 18s.

### Mitochondrial membrane potential measurement

Cells were stained with 5 nM of TMRE (Life Technology) and counter stained with Hoechst 33342. Images were obtained on a Zeiss AxioObserver.Z1 microscope and analyzed with ImageJ software (NIH).

### Statistical analysis

All data are expressed as mean ± SEM. Statistical significance was assessed with two-tailed unpaired *t*-test for two sample comparison or with ANOVA for data with more than two groups. *Post hoc* Tukey's test was performed for multiple-group comparison if ANOVA reached statistical significance. Kolmogorov–Smirnov test was used to test for normal distribution. Levene's test was used to evaluate equal variance among groups. A *P*-value of < 0.05 was considered statistically significant. Based on our previous experience with similar studies, we estimated at least 4–6 animals per group is needed to detect significant functional difference. However, the sample size was not pre-determined. Exact *P*-values are indicated in the figure legends or in Appendix Table S3.

## The paper explained

### Problem

Ischemic heart disease and heart failure are the leading causes of death in the world, and effective treatments for these diseases are lacking. Iron is linked to reactive oxygen species production, but studies investigating iron chelation as a therapy for these diseases have yielded conflicting results. It is also unclear whether a more targeted approach of modulating iron in subcellular compartments can be protective against these diseases.

### Results

We observed an increase in cardiac mitochondrial iron in mice after ischemia/reperfusion (I/R) injury. Genetic modulation of mitochondrial iron by overexpressing a mitochondrial iron export protein protected the mice against I/R injury. Additionally, mice treated with an iron chelator that is accessible to mitochondria were also protected against I/R injury. The protective effects of iron modulation were associated with reduced ROS production.

### Impact

Our results suggest that modulation of mitochondrial iron can be a therapeutic approach against ischemic heart disease and heart failure, and highlight the need for developing more targeted iron chelators.

**Expanded View** for this article is available online.

## Acknowledgements

The authors would like to thank the members of the Feinberg Cardiovascular Research Institute for their generous support. This work was supported by the Northwestern University Interdepartmental ImmunoBiology Flow Cytometry Core Facility, the Northwestern University Mouse Histology and Phenotyping Laboratory and a Cancer Center Support Grant (NCI CA060553). H.A. is supported by the NIH grants (K02 HL107448, R01 HL104181, and 1PO1 HL108795). H.-C.C. is supported by American Heart Association Predoctoral Training Grant 12PRE12030002 and a NIH T32 Training Grant (T32GM008152) given to Northwestern University.

## Author contributions

H-CC and HA designed experiments and wrote the paper; H-CC, RW, MS, TS, CC, JSS, and TL performed experiments; SVNP provided samples and reagents; AT, JSS, and KTS made conceptual and editorial contributions.

## Conflict of interest

H.A. receives speaking honoraria from Merck and is a consultant to Gerson Lehrman Group.

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
