## [Review Process File · EMBO Molecular Medicine]

Manuscript EMM-2015-05748

Reduction in mitochondrial iron alleviates cardiac damage during injury

Hsiang-Chun Chang, Rongxue Wu, Meng Shang, Tatsuya Sato, Chunlei Chen, Jason S. Shapiro, Ting Liu, Anita Thakur, Konrad T. Sawicki, Sathyamangla V. Naga Prasad, and Hossein Ardehali

Corresponding author: Hossein Ardehali, Northwestern University

Review timeline:

Submission date:	13 August 2015
Editorial Decision:	28 August 2015
Revision received:	18 December 2015
Editorial Decision:	11 January 2016
Revision received:	18 January 2016
Accepted:	19 January 2016

Transaction Report:

Editor: Roberto Buccione

1st Editorial Decision

28 August 2015

Thank you for the submission of your manuscript to EMBO Molecular Medicine. We have now heard back from the three Reviewers whom we asked to evaluate your manuscript.

As you will see the issues raised are few but fundamental. Although I will not dwell into much detail, I would like to highlight the main points.

Reviewer 2 is positive, but raises a few important issues. These include in general the request for improved presentation of experimental details and discussion of the literature. I should add that both points are a leitmotif throughout the three evaluations. Other, more specific concerns include the echocardiography methodology and unclear evidence of reperfusion.

Reviewer 3 mentions several specific items of concern including on mitochondrial purity, assessment of labile iron, and the chelator treatments.

Reviewer 1 is clearly much more reserved although, I admit, perhaps somewhat cursory in his/her comments. However, s/he does raise some very important points that impinge on the overall conclusiveness and impact of your work, including dissatisfaction with data and experimental presentation and discussion of the literature (as do the other Reviewers). After further consultation among the Reviewers and in depth internal discussion, it was agreed that evidence for mitochondrial functionality and activity should be provided. The interest of this manuscript is an eminently translational one, considering that the specific conceptual advance is limited by previous reports and that the mechanistic angle is not especially explored. For this reason, the provided evidence should

be complete and solid to make it of broad interest beyond a specialist readership.

In conclusion, while publication of the paper cannot be considered at this stage, given the potential interest of your findings and after internal discussion, we have decided to give you the opportunity to address the above concerns.

We are thus prepared to consider a substantially revised submission, with the understanding that the Reviewers' concerns must be addressed with additional experimental data where appropriate and that acceptance of the manuscript will entail a second round of review. The overall aim is to significantly upgrade the relevance and usefulness of the dataset, which of course is of paramount importance for our title.

I understand that if you do not have the required data available at least in part, to address the above, this might entail a significant amount of time, additional work and experimentation and might be technically challenging, I would therefore understand if you chose to rather seek publication elsewhere at this stage. Should you do so, we would welcome a message to this effect.

Please note that it is EMBO Molecular Medicine policy to allow a single round of revision only and that, therefore, acceptance or rejection of the manuscript will depend on the completeness of your responses included in the next, final version of the manuscript.

EMBO Molecular Medicine now requires a complete author checklist (<http://embomolmed.embopress.org/authorguide#editorial3>) to be submitted with all revised manuscripts. Provision of the author checklist is mandatory at revision stage; The checklist is designed to enhance and standardize reporting of key information in research papers and to support reanalysis and repetition of experiments by the community. The list covers key information for figure panels and captions and focuses on statistics, the reporting of reagents, animal models and human subject-derived data, as well as guidance to optimise data accessibility.

As you know, EMBO Molecular Medicine has a "scooping protection" policy, whereby similar findings that are published by others during review or revision are not a criterion for rejection. However, I do ask you to get in touch with us after three months if you have not completed your revision, to update us on the status. Please also contact us as soon as possible if similar work is published elsewhere.

***** Reviewer's comments *****

Referee #1 (Comments on Novelty/Model System):

The role of mitochondrial damage and oxidative stress induced cellular death already established in human diseases including cardiac ischemia/reperfusion. In addition the role of heavy metal such as iron involvement in the pathobiology of cardiac ischemia reperfusion becomes one of hottest area of the research. Because excess cellular iron increases reactive oxygen species (ROS) production and causes cellular damage especially energetic machine. Mitochondria are the major site of iron metabolism and ROS production; however, few studies investigated the role of mitochondrial iron in the development of cardiac disorders, such as ischemic heart disease or cardiomyopathy (CM). Authors' of this paper attempted to determine how the increased mitochondrial iron in mice after ischemia/reperfusion (I/R) and in human hearts with ischemic CM, and hypothesize that decreasing mitochondrial iron protects against I/R damage and the development of CM. Reducing mitochondrial iron genetically though cardiac-specific overexpression of a mitochondrial iron export protein or pharmacologically using a mitochondria-permeable iron chelator (rather than iron chelators that cannot penetrate the mitochondria) protects mice against I/R injury. Furthermore, decreasing mitochondrial iron protects the murine hearts in a model of spontaneous CM with mitochondrial iron accumulation. Reduced mitochondrial ROS that is independent of alterations in the electron transport chain ROS production capacity contributes to the protective effects. Based on this observation authors' conclusion is that mitochondrial iron contributes to cardiac ischemic damage, and may be a novel therapeutic target against ischemic heart disease.

I am very impressed with the authors' hypothesis and goal of this study. Unfortunately, at the present conditions in my judgment, this manuscript has several key weaknesses. Consequently, validating authors' observations and significance is difficult without addressing some of the following issues:

1. Study design and experimental tasks utilized by the authors have some poor points. Authors' should describe pattern of the mitochondrial dynamics during the ischemia/ reperfusion as well as when mitochondria-permeable iron chelator treatment.
2. Authors' should provide cytological pattern of the changes seen in selected markers overexpression and potential quantitative results from this experiments that would be necessary for the interpretation of their results and validation of the changes that seen in an different experimental conditions.
3. Major problems with this manuscript are that it is very difficult to make judgment the regarding the conclusion that based on the authors' observation. Authors' should describe detailed features of the mitochondrial complex I-V that involved in this effect.
4. It has been already demonstrated that ischemia/reperfusion has direct effects on the mitochondrial membrane potential and future mitochondrial ability to provide energy in ischemia/reperfusion as well as other diseases conditions. Without studying mitochondrial membrane potential it is almost impossible to validate significance of authors' observations.
5. Based on the literature evidence toxic action of the free radicals most likely plays pivotal roles on this type pathology. Therefore authors should search literature evidence regarding the results observed in this current study.
6. Authors' should determine pattern of the nitric oxide synthase pattern in their experiments. Without these experiments current study does not make any scientific sense. Because especially in the cases of the mitochondria Nitric oxide involvement appeared to be one of primary and key component that initiate mitochondrial functional failure and structural lesions.
7. Terse but effective language should be used in order to make much clear the results and discussion without removing content central to the investigation.
8. Literature search also very weak and often does not make any sense. I strongly recommend pubmed.com search by using key words.
9. Overall these manuscripts repeat many of recently published results and therefore it is impossible to validate significance and novelty of the current study.

Therefore at the present form this manuscript cannot be considered as a completed study.

Referee #1 (Remarks):

The role of mitochondrial damage and oxidative stress induced cellular death already established in human diseases including cardiac ischemia/reperfusion. In addition the role of heavy metal such as iron involvement in the pathobiology of cardiac ischemia reperfusion becomes one of hottest area of the research. Because excess cellular iron increases reactive oxygen species (ROS) production and causes cellular damage especially energetic machine. Mitochondria are the major site of iron metabolism and ROS production; however, few studies investigated the role of mitochondrial iron in the development of cardiac disorders, such as ischemic heart disease or cardiomyopathy (CM). Authors' of this paper attempted to determine how the increased mitochondrial iron in mice after ischemia/reperfusion (I/R) and in human hearts with ischemic CM, and hypothesize that decreasing mitochondrial iron protects against I/R damage and the development of CM. Reducing mitochondrial iron genetically though cardiac-specific overexpression of a mitochondrial iron export protein or pharmacologically using a mitochondria-permeable iron chelator (rather than iron chelators that cannot penetrate the mitochondria) protects mice against I/R injury. Furthermore, decreasing mitochondrial iron protects the murine hearts in a model of spontaneous CM with mitochondrial iron accumulation. Reduced mitochondrial ROS that is independent of alterations in the electron transport chain ROS production capacity contributes to the protective effects. Based on this observation authors' conclusion is that mitochondrial iron contributes to cardiac ischemic damage, and may be a novel therapeutic target against ischemic heart disease.

I am very impressed with the authors' hypothesis and goal of this study. Unfortunately, at the present conditions in my judgment, this manuscript has several key weaknesses. Consequently, validating authors' observations and significance is difficult without addressing some of the following issues:

1. Study design and experimental tasks utilized by the authors have some poor points. Authors' should describe pattern of the mitochondrial dynamics during the ischemia/ reperfusion as well as when mitochondria-permeable iron chelator treatment.
2. Authors' should provide cytological pattern of the changes seen in selected markers overexpression and potential quantitative results from this experiments that would be necessary for the interpretation of their results and validation of the changes that seen in an different experimental conditions.
3. Major problems with this manuscript are that it is very difficult to make judgment the regarding the conclusion that based on the authors' observation. Authors' should describe detailed features of the mitochondrial complex I-V that involved in this effect.
4. It has been already demonstrated that ischemia/reperfusion has direct effects on the mitochondrial membrane potential and future mitochondrial ability to provide energy in ischemia/reperfusion as well as other diseases conditions. Without studying mitochondrial membrane potential it is almost impossible to validate significance of authors' observations.
5. Based on the literature evidence toxic action of the free radicals most likely plays pivotal roles on this type pathology. Therefore authors should search literature evidence regarding the results observed in this current study.
6. Authors' should determine pattern of the nitric oxide synthase pattern in their experiments. Without these experiments current study does not make any scientific sense. Because especially in the cases of the mitochondria Nitric oxide involvement appeared to be one of primary and key component that initiate mitochondrial functional failure and structural lesions.
7. Terse but effective language should be used in order to make much clear the results and discussion without removing content central to the investigation.
8. Literature search also very weak and often does not make any sense. I strongly recommend pubmed.com search by using key words.
9. Overall these manuscripts repeat many of recently published results and therefore it is impossible to validate significance and novelty of the current study.

Therefore at the present form this manuscript cannot be considered as a completed study.

Referee #2 (Remarks):

Chang and colleagues addressed the role of iron overload in the context of myocardial ischemia-reperfusion injury. The study contains data showing an increase post-ischemic mitochondrial iron content. Overexpression of a mitochondrial iron transporter (which decreases mitochondrial iron content) reduces post-ischemic death and dysfunction. Likewise, deletion of the same transporter exacerbates post-ischemic mitochondrial iron overload and ventricular dysfunction. The mitochondrial permeable iron chelator, BPD, rescued the pathologic phenotype of the iron transporter deficient mouse.

The manuscript is clearly written and the experimental design is robust. There are, however, a few points to be addressed, which would improve the readers' experience with this study.

General:

1. The infarcts appear large and transmural, which is more akin to the expectation of a non-reperfused model's infarcts. What confirmation do the authors have that the coronaries are actually reperfused? Is it possible that there were simply differences in post-ischemic arterial patency (perhaps no-reflow)?
2. If the ABCB8 Tg mice favor mitochondrial iron efflux, where does the iron go (particularly in the post-ischemic setting)? According to Figure 3, it does not show up in the cytosol? At minimum, please address this in the Discussion.

Specific:

1. Although ABCB8 transgenic and knockout mice are used, no data are provided to demonstrate the gain/loss of expression. Please include such data; including it in the supplement is acceptable.

- Likewise, please mention in the Methods the cre-driver transgenic mouse that was used.
2. Figure 5C: please clarify how this was done. Is the reader simply seeing data for "scar size"? Or, is this interstitial fibrosis?
 3. The echocardiography methodology could be somewhat clearer in terms of the specific views used to generate the data shown in the paper. For example, were the EF data derived from Simpson's?
 4. This reviewer disregarded the units for "non-heme iron data" because it is not clear that those units make sense. Please clarify and/or check the calculations that generated the units in Figures 1A, 1B, 1E, 1F, 2A, 2B, 3A, 3B, etc...
 5. Considering the similar focus of work by John Eaton, the authors should consider discussing some of his group's studies (e.g. J Biol Chem. 2008. 284, 4767-4775.).
 6. Presenting the data as mean +/- SD is more meaningful to the reader than mean +/- SEM; however, I leave this to the Journal and authors to determine.
 7. It might help the reader if the authors added a section to the Discussion addressing the sources of iron that are overloading the post-ischemic mitochondria.

Referee #3 (Comments on Novelty/Model System):

There are several model systems and complementary approaches

Referee #3 (Remarks):

This work investigates the effects of mitochondrial iron manipulations in cardiac function during ischemic heart disease or cardiomyopathy. By a series of pharmacological and genetic/molecular approaches the authors conclude that reduction of mitochondrial iron overload in the heart during injury improves cardiac function, mitigates cardiac damage, and offers therapeutic benefits. The data suggest that the use of iron chelators that are permeable to mitochondria can have a translational potential for the treatment of ischemia/reperfusion (I/R) injury or cardiomyopathy.

The major strength of the work are the pathophysiological data on cardiac function, which are convincing by being validated in various cell and mouse models. There are, however, several methodological and technical issues on the mechanistic analysis and the role of mitochondrial iron; they require attention since they may affect data interpretation.

A key experimental finding is that I/R injury causes mitochondrial (but not cytosolic) iron accumulation. It appears that mitochondrial and cytosolic iron were measured following biochemical fractionation. However, there is no indication on the purity of these fractions (neither in mouse cell/tissues, nor in human tissue samples). Purity should be controlled by measuring the expression of mitochondrial and cytosolic markers (for instance, porin and IRP1). Mitochondrial purity should also be controlled in the experiments with ABCB8 transgenic mice (Fig. 3), chelator-treated wt mice (Fig. 6) and ABCB8 KO mice (Fig. 7).

Description of experimental conditions is generally poor. For how long and how much H₂O₂ was used in the experiment in Figs 1C-D, 2E-F and 8F?

For how long were the treatments with BPD and DFO in Fig. 2? If the chelator pre-treatments were short (<2 h), the weak DFO protection against H₂O₂-induced cell death could be due to the low rate of cell permeation by this drug. Previous literature suggests that DFO does protect against H₂O₂-induced cell death in various cell types, and the efficiency depends on the time of treatment (Walker & Shah *Kidney Int* 1991; Doulias et al, *FRBM* 2003; Lim et al, *Mol Pharmacol* 2008).

There is a major methodological problem with the assessment of labile iron. Again, the description is poor and crucial details (and references) are missing. The use of fluorescent chelators for the assessment of labile iron in the cytosol and mitochondria has been developed and validated by Cabantchick and coworkers (calcein assay) and by Rauen and coworkers (RPA assay), respectively. In both approaches, iron binding quenches fluorescence of the dye and quantification is made

following de-quenching with a strong cell-permeable iron chelator (SIH or PIH). Measurement of fluorescence difference is critical for iron specificity. It appears that the authors have modified the previous protocols and should therefore provide evidence for the analytical performance and accuracy of their method.

In Fig. 6, the chelator treatments are quite long (3 weeks) and may affect erythropoiesis and systemic iron metabolism. This should be controlled by measuring hematological and serum iron parameters.

Minor: The legend to Fig. 8 should be updated (description to panel A is missing).

1st Revision - authors' response

18 December 2015

We would like to thank the editor and reviewers for carefully considering our submission, and for their generally positive view of our manuscript. In response to the very helpful suggestions provided by the reviewers, we performed eleven new experiments which greatly strengthened our data and conclusions. These experiments included data on mitochondrial dynamics, mitochondrial complex activity, mitochondrial membrane potential, and NOS levels, among others. We also modified the introduction and discussion according to the reviewers' suggestions. We feel confident that our new submission provides the data and supporting discussion to convincingly demonstrate a role for mitochondrial iron in cardiac damage in response to ischemic injury. Please find our point-by-point responses below:

Referee #1 (Comments on Novelty/Model System):

I am very impressed with the authors' hypothesis and goal of this study. Unfortunately, at the present conditions in my judgment, this manuscript has several key weaknesses. Consequently, validating authors' observations and significance is difficult without addressing some of the following issues:

Study design and experimental tasks utilized by the authors have some poor points. Authors' should describe pattern of the mitochondrial dynamics during the ischemia/ reperfusion as well as when mitochondria-permeable iron chelator treatment.

This is an important point. To address the reviewer's comment, we have measured mitochondrial DNA content in cultured H9c2 cells with either ABCB8 overexpression or after chelator treatment. We have also measured expression of genes associated with mitochondrial fusion and fission and mitochondrial biogenesis. None of these parameters were different in cells with ABCB8 overexpression or with iron chelator treatment. These results are included in Supplementary Figure S7. Similar measurements were done in ABCB8 transgenic (TG) mice and chelator-treated wild-type mice (Supplementary Figure S8). Consistent with the in vitro studies, our experiments in mice did not show a change in the expression of genes involved in mitochondrial dynamics and biogenesis with mitochondrial iron modulation. Additionally, we have evaluated genes associated with mitochondrial dynamics and biogenesis in non-transgenic (NTG) and ABCB8 TG mice two days after I/R. I/R resulted in significant but comparable decrease in the expression of genes associated with mitochondrial dynamics and biogenesis in ABCB8 TG and NTG hearts. The results are included in the Supplementary Figure S9.

Authors' should provide cytological pattern of the changes seen in selected markers overexpression and potential quantitative results from this experiments that would be necessary for the interpretation of their results and validation of the changes that seen in an different experimental conditions.

We have performed immunofluorescence staining for ABCB8 in the NTG and ABCB8 TG mouse hearts and demonstrated colocalization of the overexpressed ABCB8 with the mitochondria marker Cox4. Representative micrograph is shown in Supplementary Figure S3C. We hope this experiment addresses the Reviewer's comment.

Major problems with this manuscript are that it is very difficult to make judgment regarding the conclusion that based on the authors' observation. Authors should describe detailed features of the mitochondrial complex I-V that involved in this effect.

Thank you for bringing up this excellent point. We have performed activity measurement for complex I, II and IV in ABCB8 overexpressing or chelator treated H9c2 cells with or without H₂O₂ treatment. While H₂O₂ treatment significantly inhibited the activity of mitochondrial complex I, II and IV, cells with ABCB8 overexpression or treated with BPD (but not DFO) were protected against the decrease in complex activity. The results are shown in Figure 9. Additionally, we expanded our discussion to include the contribution of changes in mitochondrial complex activity during I/R to cell death (please see page 18).

It has been already demonstrated that ischemia/reperfusion has direct effects on the mitochondrial membrane potential and future mitochondrial ability to provide energy in ischemia/reperfusion as well as other diseases conditions. Without studying mitochondrial membrane potential it is almost impossible to validate significance of authors' observations.

We agree with this comment and have measured mitochondrial membrane potential in H9c2 cells treated with iron chelators with or without H₂O₂. While chelator treatment did not influence mitochondrial membrane potential at baseline, BPD (but not DFO) pretreatment protected against the H₂O₂-mediated decrease in mitochondrial membrane potential. The results are shown in Figure 8G.

Based on the literature evidence toxic action of the free radicals most likely plays pivotal roles on this type pathology. Therefore authors should search literature evidence regarding the results observed in this current study.

Thank you for your comments. We have expanded our introduction and discussion sections to include relevant mechanisms for ROS-induced release of labile iron, including increased delivery of iron into mitochondria, damage of iron/sulfur clusters, and disruption of mitochondrial iron homeostasis (please see page 17). We also discussed that iron and associated changes in ROS can further damage mitochondrial membrane, respiratory chain complexes, and mitochondrial DNA, and contribute to cell death during I/R (please see page 3, 4 and 18).

Authors should determine pattern of the nitric oxide synthase pattern in their experiments. Without these experiments current study does not make any scientific sense. Because especially in the cases of the mitochondria Nitric oxide involvement appeared to be one of primary and key component that initiate mitochondrial functional failure and structural lesions.

We have measured the expression levels of eNOS and iNOS in our ABCB8 TG mice at baseline and after I/R. nNOS expression was below detection limit in our tissue sample. Additionally, as changes in BH₄ biosynthesis can influence NOS coupling and ROS production, we also measured genes implicated in BH₄ biosynthesis in NTG and ABCB8 TG mice at baseline and after I/R, and showed that ABCB8 overexpression was not associated with changes in the expression of these genes. The results are now included in Supplementary Figure S10. Therefore, while altered expression and uncoupling of NOS has been implicated in the structural and functional changes in mitochondria, modulation of mitochondrial iron does not appear to influence NOS-mediated cellular injury. Rather, we believe that the major protective effect of mitochondrial iron modulation is mediated through alteration in iron-catalyzed ROS formation and associated damage of mitochondrial membrane and respiratory chain complex.

Terse but effective language should be used in order to make much clear the results and discussion without removing content central to the investigation.

Thank you for your suggestions. We have now revised the manuscript to remove redundant points in the introduction and discussion sections, and to make our discussion more clear in those sections.

Literature search also very weak and often does not make any sense. I strongly recommend pubmed.com search by using key words.

Thank you for your suggestions. We have performed Medline search using the keyword “iron and heart,” “iron and ischemia/reperfusion,” “mitochondrial iron AND ROS”, and now included 14 new references in our manuscript. These references include a review by Dixon and Stockwell on the mechanism by which iron-catalyzed ROS production can cause cell death, several papers and reviews by Eaton’s group as well as a paper by Doulias et al. on the mechanism of iron mediated cell death. We also compared our findings to the review written by Lesnefsky and Hoppel on the role of mitochondria in ischemia/reperfusion injury in the aged heart. Additionally, we also referenced the paper by Zhang and Lemasters in our discussion on the potential mechanism of mitochondrial iron overload during ischemia/reperfusion. We hope that this expanded literature search and discussion provides a better and satisfactory context for the significance of our paper.

Overall these manuscripts repeat many of recently published results and therefore it is impossible to validate significance and novelty of the current study.

Thank you for your comments. We agree that cardiomyocyte iron has been studied in the heart; however, the results have been conflicting and the role of mitochondrial iron in cardiac response to ischemic injury has not been thoroughly investigated. Among the novel studies in our paper, we would like to highlight that we identified baseline mitochondrial iron as a major source of ROS damage during I/R, which can be targeted to attenuate tissue damage. Additionally, we demonstrated the beneficial effects of decreasing mitochondrial iron in I/R using both genetic and pharmacological approaches. This study provides proof-of-concept for further investigation into mitochondria-targeted iron chelators as potential therapies for ischemic heart disease and heart failure. Our study also suggests that while iron supplementation may help alleviate symptoms of heart failure patients, they may ultimately contribute to tissue damage during I/R.

Referee #2 (Remarks):

Chang and colleagues addressed the role of iron overload in the context of myocardial ischemia-reperfusion injury. The study contains data showing an increase post-ischemic mitochondrial iron content. Overexpression of a mitochondrial iron transporter (which decreases mitochondrial iron content) reduces post-ischemic death and dysfunction. Likewise, deletion of the same transporter exacerbates post-ischemic mitochondrial iron overload and ventricular dysfunction. The mitochondrial permeable iron chelator, BPD, rescued the pathologic phenotype of the iron transporter deficient mouse.

The manuscript is clearly written and the experimental design is robust. There are, however, a few points to be addressed, which would improve the readers' experience with this study.

General:

The infarcts appear large and transmural, which is more akin to the expectation of a non-reperfused model's infarcts. What confirmation do the authors have that the coronaries are actually reperfused? Is it possible that there were simply differences in post-ischemic arterial patency (perhaps no-reflow)?

Thank you for your suggestions. We adopted a protocol of 50-minute coronary occlusion to induce a relatively large myocardial injury in order to better assess the role of mitochondrial iron in cardiac ischemic injury. Thus, long coronary occlusion time was intentionally chosen in our studies.

Nevertheless, reperfusion of myocardium was visually verified by the normalization of myocardium color. Additionally, we have performed Evan’s blue staining to demonstrate the patency of the reopened coronary artery in ABCB8 TG and NTG mice. Briefly, 24-hour after I/R, mice were anesthetized and Evan’s blue were injected into aortic root without re-ligation of the coronary artery. The hearts were then excised and sections below the original ligature was photographed. As show in the pictures below, the entire myocardium was stained blue, demonstrating patency of the coronary artery after I/R.

NTG

ABCB8 TG

If the ABCB8 Tag mice favor mitochondrial iron efflux, where does the iron go (particularly in the post-ischemic setting)? According to Figure 3, it does not show up in the cytosol? At minimum, please address this in the Discussion.

Thank you for your comment. Mitochondrial iron only represents a small fraction of total cellular iron, as evidenced by the distribution of radioactive iron in cytosolic and mitochondrial fractions from cells incubated with ^{55}Fe . Therefore, shifting a small fraction of iron from mitochondria into cytosolic iron pool may not have a great impact on total cytosolic iron amount. We have also included this explanation in the discussion section.

Specific:

Although ABCB8 transgenic and knockout mice are used, no data are provided to demonstrate the gain/loss of expression. Please include such data; including it in the supplement is acceptable. Likewise, please mention in the Methods the cre-driver transgenic mouse that was used.

Thank you for this comment. We have included protein levels and mRNA expression of ABCB8 in TG mice in Supplementary Figure S3 and the ABCB8 protein levels in knockout mice in Supplementary Figure S6. Additionally, we have revised the method section to provide information on the α -MHC MER-Cre-MER transgenic mice that were used to generate cardiac-specific knockout of ABCB8.

Figure 5C: please clarify how this was done. Is the reader simply seeing data for "scar size"? Or, is this interstitial fibrosis?

The infarct size is calculated as the arc length of fibrotic scar over the circumference of the left ventricle. This information is now included in the method section.

The echocardiography methodology could be somewhat clearer in terms of the specific views used to generate the data shown in the paper. For example, were the EF data derived from Simpson's?

M-mode images obtained from long and short axis views were used to calculate the EF. To reflect the changes in systolic function after injury, M mode images were acquired in areas with visible impairment of contraction after I/R. The machine calculated ejection fraction was based on Teichholtz equation. This information has been added to the method section. Additionally, we now

include fractional shortening, which was based on end-systolic and end-diastolic left ventricle dimensions on M-mode images (Fig 3D, Supplementary Figure S4-5).

This reviewer disregarded the units for "non-heme iron data" because it is not clear that those units make sense. Please clarify and/or check the calculations the generated the units in Figures 1A, 1B, 1E, 1F, 2A, 2B, 3A, 3B, etc...

For radioactive iron measurement, ^{55}Fe radioactivity (cpm) was normalized to the protein concentration of each sample. We have also updated the units for the colorimetric method to $\mu\text{mole/g}$ protein for other figures to avoid confusion.

Considering the similar focus of work by John Eaton, the authors should consider discussing some of his group's studies (e.g. *J Biol Chem.* 2008. 284, 4767-4775.).

Thank you for your suggestion. We have now included relevant literature from the group. Their findings on the mechanism for mitochondrial damage in cells with iron overload (Eaton & Qian, 2002; Gao et al, 2009) were reviewed in the introduction. Additionally, their studies on lysosomal iron mediating H_2O_2 - or radiation-induced cell death (Kurz et al, 2010; Persson et al, 2005; Yu et al, 2003) were also discussed in the introduction.

Presenting the data as mean \pm SD is more meaningful to the reader than mean \pm SEM; however, I leave this to the Journal and authors to determine.

Thank you. If the Reviewer and the Journal agree, we would like to leave it the data as mean \pm SEM, since changing all of the graphs will be time-consuming. If it is decided that mean \pm SD is necessary, we would be happy to do that.

It might help the reader if the authors added a section to the Discussion addressing the sources of iron that are overloading the post-ischemic mitochondria.

Thank you for your suggestion. We have now included the following paragraph in the discussion section:

In I/R, in which ROS levels are increased, the amount of free iron in the mitochondria may be higher than basal conditions. Increased labile iron may also disrupt mitochondrial iron homeostasis, resulting in the increased mitochondrial iron observed in the acute phase of I/R injury in our in vivo studies. Furthermore, increased lysosomal delivery of iron to mitochondria during ischemia can also lead to increased mitochondrial iron levels (Zhang & Lemasters, 2013). It is possible that all of these mechanisms contribute to the increase in mitochondrial iron in mouse hearts after I/R.

Referee #3 (Remarks):

This work investigates the effects of mitochondrial iron manipulations in cardiac function during ischemic heart disease or cardiomyopathy. By a series of pharmacological and genetic/molecular approaches the authors conclude that reduction of mitochondrial iron overload in the heart during injury improves cardiac function, mitigates cardiac damage, and offers therapeutic benefits. The data suggest that the use of iron chelators that are permeable to mitochondria can have a translational potential for the treatment of ischemia/reperfusion (I/R) injury or cardiomyopathy.

The major strength of the work are the pathophysiological data on cardiac function, which are convincing by being validated in various cell and mouse models. There are, however, several methodological and technical issues on the mechanistic analysis and the role of mitochondrial iron; they require attention since they may affect data interpretation.

A key experimental finding is that I/R injury causes mitochondrial (but not cytosolic) iron accumulation. It appears that mitochondrial and cytosolic iron were measured following

biochemical fractionation. However, there is no indication on the purity of these fractions (neither in mouse cell/tissues, nor in human tissue samples). Purity should be controlled by measuring the expression of mitochondrial and cytosolic markers (for instance, porin and IRP1). Mitochondrial purity should also be controlled in the experiments with ABCB8 transgenic mice (Fig. 3), chelator-treated wt mice (Fig. 6) and ABCB8 KO mice (Fig. 7).

We have now included western blotting data to demonstrate the purity of purified subcellular fractions (Supplementary Fig. S1).

Description of experimental conditions is generally poor. For how long and how much H₂O₂ was used in the experiment in Figs 1C-D, 2E-F and 8F?

Thank you for pointing this out. Cells were treated with 600 μM H₂O₂ for 6 hours in those experiments. We have now included experimental details in the methods section.

For how long were the treatments with BPD and DFO in Fig. 2? If the chelator pre-treatments were short (<2 h), the weak DFO protection against H₂O₂-induced cell death could be due to the low rate of cell permeation by this drug. Previous literature suggests that DFO does protect against H₂O₂-induced cell death in various cell types, and the efficiency depends on the time of treatment (Walker & Shah *Kidney Int* 1991; Doulias et al, *FRBM* 2003; Lim et al, *Mol Pharmacol* 2008).

Cells were pre-treated with BPD or DFO for 2 hours, and this information is now included in the methods section. We agree that the prolonged treatment of DFO would deplete cellular iron, and eventually could result in lower mitochondrial iron *in vitro*. This change in mitochondrial iron in cells with long DFO treatment can certainly explain some of the protective effects seen in the papers that the reviewer cited. However, in an *in vivo* setting, it would require significant depletion of circulating iron to achieve the protective effect seen in those *in vitro* studies. The purpose of our study is to highlight that by using an iron chelator that can modulate mitochondrial iron, we can confer cardioprotection without significantly impairing cardiac function at baseline.

There is a major methodological problem with the assessment of labile iron. Again, the description is poor and crucial details (and references) are missing. The use of fluorescent chelators for the assessment of labile iron in the cytosol and mitochondria has been developed and validated by Cabantchick and coworkers (calcein assay) and by Rauen and coworkers (RPA assay), respectively. In both approaches, iron binding quenches fluorescence of the dye and quantification is made following de-quenching with a strong cell-permeable iron chelator (SIH or PIH). Measurement of fluorescence difference is critical for iron specificity. It appears that the authors have modified the previous protocols and should therefore provide evidence for the analytical performance and accuracy of their method.

We have now performed additional measurements using methods by Cabantchick et al. and Rauen et al. The amount of labile iron was quantified using the difference between the baseline fluorescence measurement and the fluorescence signal after PIH treatment, which was converted to iron concentration using a standard curve generated with increasing concentrations of fluorescent indicators. Lastly, to account for the difference in cell number during plating, Hoechst 33342 and mitoTracker green signal was used for normalization of cytosolic and mitochondrial labile iron, respectively. Figures 1C-D, 2D and 2F have been updated to reflect the change.

In Fig. 6, the chelator treatments are quite long (3 weeks) and may affect erythropoiesis and systemic iron metabolism. This should be controlled by measuring hematological and serum iron parameters.

We have measured systemic iron parameters and performed complete blood counts after three weeks of iron chelator treatment. While serum iron was significantly decreased after chelator treatment, we did not see significant changes in erythropoiesis. The results are shown in Supplementary Table S1.

Minor: The legend to Fig. 8 should be updated (description to panel A is missing).

Thank you. The figure legend has been updated.

2nd Editorial Decision

11 January 2016

Thank you for the submission of your manuscript to EMBO Molecular Medicine. We have now heard back from the three Reviewers whom we asked to evaluate your manuscript.

As you will see the Reviewers are now satisfied with your manuscript and I am thus prepared to accept your manuscript for publication pending the following editorial amendments:

- 1) Please note the comment by Reviewer 3
- 2) As per our Author Guidelines, the description of all reported data that includes statistical testing must state the name of the statistical test used to generate error bars and P values, the number (n) of independent experiments underlying each data point (not replicate measures of one sample), and the actual P value for each test (not merely 'significant' or 'P < 0.05').
- 3) The manuscript must include a statement in the Materials and Methods identifying the institutional and/or licensing committee approving the experiments, including any relevant details (like how many animals were used, of which gender, at what age, which strains, if genetically modified, on which background, housing details, etc). I note that you have not provided all this information in your manuscript. We encourage authors to follow the ARRIVE guidelines for reporting studies involving animals. Please see the EQUATOR website for details: <http://www.equator-network.org/reporting-guidelines/improving-bioscience-research-reporting-the-arrive-guidelines-for-reporting-animal-research/>. Please make sure that all the above details are reported.
- 4) I would also like to mention that in my previous decision letter I had asked you to provide a checklist (<http://embomolmed.embopress.org/authorguide#editorial3>) which is mandatory with all revised manuscripts. Please do so with your next version of your manuscript. The checklist is designed to enhance and standardize reporting of key information in research papers, and to support reanalysis and repetition of experiments by the community. The list covers key information for figure panels and captions and focuses on statistics, the reporting of reagents, animal models and human subject-derived data (and therefore connected to my point 3 above), as well as guidance to optimise data accessibility.
- 5) Every published paper now includes a 'Synopsis' to further enhance discoverability. Synopses are displayed on the journal webpage and are freely accessible to all readers. They include a short standfirst - to be written by the editor - as well as 2-5 one sentence bullet points that summarise the paper (to be written by the author). Please provide the short list of bullet points that summarise the key NEW findings. The bullet points should be designed to be complementary to the abstract - i.e. not repeat the same text. We encourage inclusion of key acronyms and quantitative information. Please use the passive voice. Please attach these in a separate file or send them by email, we will incorporate them accordingly.
- 6) We are now encouraging the publication of source data, particularly for electrophoretic gels and blots, with the aim of making primary data more accessible and transparent to the reader. Would you be willing to provide a PDF file per figure that contains the original, uncropped and unprocessed scans of all or at least the key gels used in the manuscript? The PDF files should be labeled with the appropriate figure/panel number, and should have molecular weight markers; further annotation may be useful but is not essential. The PDF files will be published online with the article as supplementary "Source Data" files. If you have any questions regarding this just contact me.
- 7) I note that some figures are organized in a "landscape" orientation. Please re-arrange the panels in each figure into a "portrait" orientation.

I look forward to reading a new revised version of your manuscript as soon as possible.

***** Reviewer's comments *****

Referee #1 (Remarks):

Revised version of this manuscript it is acceptable.

Referee #2 (Remarks):

I have no new comments. The authors' response sufficiently addressed my previous concerns.

Referee #3 (Remarks):

All issues have been addressed. Please, correct typo on Fig. S1 (tubulin).

2nd Revision - authors' response

18 January 2016

1) Please note the comment by Reviewer 3

The typo has now been corrected.

2) As per our Author Guidelines, the description of all reported data that includes statistical testing must state the name of the statistical test used to generate error bars and P values, the number (n) of independent experiments underlying each data point (not replicate measures of one sample), and the actual P value for each test (not merely 'significant' or ' $P < 0.05$ ').

The statistical test for each panel as well as the independent samples/ mice number for each group are now included in figure legends.

3) The manuscript must include a statement in the Materials and Methods identifying the institutional and/or licensing committee approving the experiments, including any relevant details (like how many animals were used, of which gender, at what age, which strains, if genetically modified, on which background, housing details, etc). I note that you have not provided all this information in your manuscript. We encourage authors to follow the ARRIVE guidelines for reporting studies involving animals. Please see the EQUATOR website for details: <http://www.equator-network.org/reporting-guidelines/improving-bioscience-research-reporting-the-arrive-guidelines-for-reporting-animal-research/>. Please make sure that all the above details are reported.

Details related to animal studies are now added to the methods section.

4) I would also like to mention that in my previous decision letter I had asked you to provide a checklist (<http://embomolmed.embopress.org/authorguide#editorial3>) which is mandatory with all revised manuscripts. Please do so with your next version of your manuscript. The checklist is designed to enhance and standardize reporting of key information in research papers, and to support reanalysis and repetition of experiments by the community. The list covers key information for figure panels and captions and focuses on statistics, the reporting of reagents, animal models and human subject-derived data (and therefore connected to my point 3 above), as well as guidance to optimise data accessibility.

The checklist is now attached.

5) Every published paper now includes a 'Synopsis' to further enhance discoverability. Synopses are displayed on the journal webpage and are freely accessible to all readers. They include a short

standfirst - to be written by the editor - as well as 2-5 one sentence bullet points that summarise the paper (to be written by the author). Please provide the short list of bullet points that summarise the key NEW findings. The bullet points should be designed to be complementary to the abstract - i.e. not repeat the same text. We encourage inclusion of key acronyms and quantitative information. Please use the passive voice. Please attach these in a separate file or send them by email, we will incorporate them accordingly.

A separate file containing list of bullet points is attached.

6) We are now encouraging the publication of source data, particularly for electrophoretic gels and blots, with the aim of making primary data more accessible and transparent to the reader. Would you be willing to provide a PDF file per figure that contains the original, uncropped and unprocessed scans of all or at least the key gels used in the manuscript? The PDF files should be labeled with the appropriate figure/panel number, and should have molecular weight markers; further annotation may be useful but is not essential. The PDF files will be published online with the article as supplementary "Source Data" files. If you have any questions regarding this just contact me.

The uncropped gels are now included as supplementary information.

7) I note that some figures are organized in a "landscape" orientation. Please re-arrange the panels in each figure into a "portrait" orientation.

We have rearranged all the figures to portrait orientation.

Referee #3 (Remarks):

All issues have been addressed. Please, correct typo on Fig. S1 (tubulin).

We apologize for the typo. It is now corrected.